



# The Role of Contact Angle and Pore Width on Pore Condensation and Freezing

Robert O. David[1,a], Jonas Fahrni[2], Claudia Marcolli[1], Fabian Mahrt[1], Dominik Brühwiler[2], and Zamin A. Kanji[1]

[1]Institute for Atmospheric and Climate Science, ETH Zürich, 8092 Zürich, Switzerland
[2]Institute of Chemistry and Biotechnology, Zürich University of Applied Sciences (ZHAW), 8820 Wädenswil, Switzerland
[a]Now at Department of Geosciences, University of Oslo, Oslo, 0315, Norway

*Correspondence to*: Robert O. David (r.o.david@geo.uio.no) or Zamin A. Kanji (zamin.kanji@env.ethz.ch)

**Abstract.** It has recently been shown that pore condensation and freezing (PCF) is a mechanism responsible for ice formation under cirrus cloud conditions. PCF is defined as the condensation of liquid water in narrow capillaries below water saturation due to the Kelvin effect, followed by either heterogeneous or homogeneous nucleation depending on the temperature regime and presence of an ice nucleating active site. By using sol-gel synthesized silica with well-defined pore diameters, morphology and distinct chemical surface-functionalization, the role of the water-silica contact angle and pore width on PCF is investigated. We find that contact angle and pore width play an important role in determining the relative humidity required for capillary condensation as predicted by the Kelvin effect and subsequent ice nucleation at cirrus temperatures. For the pore diameters and contact angles covered in this study, $2.2 - 9.2$ nm and $15 - 78°$, respectively, our results reveal that the contact angle plays an important role in predicting the humidity required for pore filling while the pore diameter determines the ability of pore water to freeze. For $T > 235$ K and below water saturation, pore diameters and contact angles were not able to predict the freezing ability of the particles suggesting an absence of active sites, thus ice nucleation did not proceed via a PCF mechanism. Rather, the ice nucleating ability of the particles depended solely on chemical functionalization. Therefore, parameterizations for the ice nucleating abilities of particles at cirrus conditions should differ from parameterizations at mixed-phase clouds conditions. Our results support PCF as the atmospherically relevant ice nucleation mechanism below water saturation when porous surfaces are encountered in the troposphere.

## 1 Introduction

In the Earth's atmosphere, ice crystals are important for precipitation formation (Mülmenstädt et al., 2015), cloud lifetime, radiative properties, and ultimately modulate climate (McFarquhar et al., 2017; Seinfeld et al., 2016). Understanding the formation of ice crystals is therefore essential to accurately predict cloud properties and thus future climate. The freezing temperature of pure water droplets is approximately 235 K, known as the homogeneous freezing temperature (HFT). However, ice formation is also observed at temperature ($T$) > HFT or below water saturation. At $T$ > HFT ice formation takes place heterogeneously and is initiated by the presence of a so-called ice active site (Fletcher, 1969; Kaufmann et al., 2017; Kiselev





et al., 2017; Vali et al., 2015), which lowers the energy barrier required for the homogeneous nucleation of ice. Below water saturation, ice nucleation is conventionally defined as deposition nucleation, or the direct transition from water vapour to the ice phase without an intermediate liquid water phase (Pruppacher and Klett, 1997; Vali et al., 2015). However, it has been shown that ice nucleation below water saturation occurs in the presence of cracks and steps (Campbell et al., 2017; Christenson,

2013; David et al., 2019; Fukuta, 1966; Higuchi and Fukuta, 1966; Kovács et al., 2012; Kovács and Christenson, 2012; Wang et al., 2016) and has subsequently been termed pore condensation and freezing (PCF) (Campbell and Christenson, 2018; David et al., 2019; Marcolli, 2014; Vali et al., 2015; Wagner et al., 2016). PCF occurs when bulk water, which can exist below water saturation in narrow pores, cracks, cavities or capillaries (hereafter referred to as pores) freezes. Due to the concave curvature of water in confinements, the vapour pressure required for condensation to occur in a pore compared to a flat/bulk water surface

can be predicted by the inverse version of the Kelvin equation given as:

$$\frac{p_{lc}}{p_l} = exp\left[\frac{-4\gamma(T)v_l(T)}{\frac{D}{\cos\theta}RT}\right],\qquad(1)$$

where $p_{lc}$ is the vapor pressure of water over a concave surface, $p_l$ is the vapor pressure of water over a flat surface, and $\frac{p_{lc}}{p_l}$ denotes the saturation ratio w.r.t water, while $\gamma(T)$ is the temperature dependent surface tension of the water-vapour interface, $v_l(T)$ is the molar volume of water as a function of temperature, $D$ is the pore diameter, $R$ is the gas constant and $T$ is the

temperature in Kelvin. $\theta$ is the contact angle of water on the (pore) material or the wettability of the material, where $\theta = 0°$ ($\cos\theta = 1$) denotes a perfectly wettable surface, whereas higher contact angles denote less hydrophilic surfaces (Lohmann et al., 2016). As deduced from Eq. 1, the relative humidity w.r.t. water ($RH_w$) required for a pore to fill depends on the pore diameter and the contact angle of the pore surface. As such, at a given contact angle a narrower pore will fill at a lower $RH_w$ than a wider pore. Conversely, for a fixed pore diameter, the higher the contact angle of water on the pore surface, the higher

the $RH_w$ required for pore filling.

Once the pore is filled, the water can freeze either homogeneously or heterogeneously depending on the temperature regime or the presence of an ice active site, as long as the pore is wide enough to host the critical ice germ (Campbell et al., 2017; Campbell and Christenson, 2018; David et al., 2019; Koop, 2017; Marcolli, 2014). In order for the phase transition from supercooled water to ice to occur, Classical Nucleation Theory predicts that a large enough cluster of water molecules, known

as a germ, must organize into ice before the entire water volume can freeze (Fletcher, 1962; Lohmann et al., 2016; Pruppacher and Klett, 1997). The radius of this critical ice germ ($r_c$) can be calculated as:

$$r_c = \frac{2\sigma_{iw}v_{ice}}{RT\ln(S_i)},\qquad(2)$$

where $\sigma_{iw}$ is interfacial energy between the ice and water interface, $v_{ice}$ is the approximate volume of bulk ice, and $S_i$ is the supersaturation with respect to ice. Additionally, it has been shown that even down to extreme supercooling ($T < 200$ K), a

quasi-liquid layer of water is present along the pore wall (Jähnert et al., 2008; Marcolli, 2014; Moore et al., 2010; Schreiber et al., 2001). Thus, the diameter of a pore capable of hosting ice ($D_p$) can be expressed as:





$$D_p \geq 2r_c + 2t, \tag{3}$$

where $t$ is the thickness of the quasi-liquid layer, assumed as $t = 0.38$ nm (Schreiber et al., 2001). Indeed, Marcolli (2014) reported that $D_p$ is a good predictor for ice forming in porous silica particles. However, once pore ice is formed, it must grow out of the pore, i.e. into the unconfined vapour region. Based on CNT the energy cost for ice growth into the vapour phase is more expensive than for growth within water. This increased cost comes from the need to replace, $\sigma_{iw}$ with $\sigma_{iv}$ in Eq. 2, which is approximately a factor of 4.8 larger than $\sigma_{iw}$ at 236 K (Cooper, 1974; Ickes et al., 2015; Ketcham and Hobbs, 1969). Additionally, the ice forming in a pore experiences water saturation and therefore the $S_i$ in Eq. 2 is higher than the $S_i$ outside of the pore. Therefore, the critical radius for ice growth out of the pore is much larger than that of the critical radius in the pore necessitating a substantial increase in $S_i$ for ice to be able to grow out of a pore (David et al., 2019; Koop, 2017). Indeed, Campbell et al, (2017) and Campbell and Christenson (2018), showed that an increase in supersaturation is required for crystallites formed in wedge shaped pores to emerge into the unconfined vapour region, which they interpreted as a second energy barrier for ice growth out of pores. In addition, molecular dynamic simulations (MDS) conducted by Page and Sear (2006) showed that protein crystal nucleation out of single pores is maximized when the pore width is close to the critical nucleus size in order to minimize the energy for pore filling and for the crystal growth out of the pore. Conversely, mesoporous silica with closely spaced cylindrical pores did not reveal any inhibition of ice growth out of pores (David et al., 2019). This result is supported through MDS and CNT based calculations revealing that an arrangement of several subcritical cylindrical pores closely spaced together greatly decreases $S_i$ required for ice growth out of pores due to pore-ice bridging across adjacent pores (David et al., 2019).

Although there is strong evidence that pores are responsible for ice nucleation below water saturation, the ability of PCF to predict ice nucleation as a function of pore width and contact angle has not been shown systematically. For example, in an earlier study we showed that pores were responsible for the observed ice nucleation of synthesized silica and NX-illite particles and that the humidities required for ice formation were consistent with PCF (David et al., 2019). Here we present results from synthesized porous silica with well-defined pore diameters, geometry, and contact angles to better understand the PCF mechanism and its predictive capability for ice nucleation at water subsaturated conditions.

## 2. Methods

### 2.1 Particle synthesis

#### 2.1.1 Synthesis of MCM-41 submicron mesoporous silica particles

Following Beck et al., (1992), $NH_4OH$ (121 mL, 28 %, Sigma-Aldrich), deionized water (300 mL) and ethanol (500 mL, 99.8 %, Sigma-Aldrich) were stirred for 5 min in a 1 L polypropylene beaker. For the synthesis of materials with 2.8 nm or





3.3 nm pores, $C_{16}$TMABr (hexadecyltrimethylammonium bromide, 1.74 g, 99 %, Acros) was subsequently added and stirred for 15 min before TEOS (tetraethoxysilane, 4.5 mL, 20.2 mmol, 98 %, Sigma-Aldrich) was quickly dropped into the reaction mixture. For 2.5 nm pores, a mixture of $C_{16}$TMABr (0.871 g, 99 %) and $C_{14}$TMABr (tetradecyltrimethylammonium bromide, 0.804 g, 99 %, Sigma-Aldrich) was used. After a few minutes, silica started to precipitate. The reaction was stirred at room temperature for 2 h before filtering (Sartorius® 393). The filter cake was subsequently washed twice with 50 mL of deionized water, dried at T = 80 ˚C for approximately 1 h and finally ground in methanol for 3 min. To obtain 3.3 nm pores, the dried particles were transferred into a Teflon bomb (Parr 4748), suspended in deionized water (80 mL), aged (80 °C , 24 h), subsequently filtered (Sartorius® 393), dried, and ground in methanol (99 %, Sigma-Aldrich). After drying again (80 ˚C, ≥ 1 h), the particles were calcined at 550 ˚C for 12 h.

### 2.1.2 Synthesis of SBA-15 submicron mesoporous silica particles

Similar to Linton et al., (2009b) Pluronic® P104 (1.25 g, BASF) was dissolved under vigorous stirring in a hydrochloric acid solution (200 mL, 1.6 mol·L$^{-1}$) at 60 ˚C and TMOS (tetramethoxysilane, 8 mL, 99 %, Sigma-Aldrich) was added quickly under vigorous stirring. After 1 min (approximate hydrolysis time, Linton et al., 2009a) the stirring rate was lowered to moderate stirring. After another 1 min, the reaction mixture was diluted with a hydrochloric acid solution (200 mL, 1.6 mol·L$^{-1}$) leading to precipitation of the silica. The reaction mixture was further stirred at 60 °C for 24 h. The resulting suspension was centrifuged, washed with deionized water (200 mL) twice, and the product was transferred to a Teflon bomb. The wet particles were dispersed in deionized water (60 mL) and the pH was adjusted to 9 by the addition of NH$_4$OH (1.05 mL, 28 %). The mixture was aged in quiescent conditions at 80 ˚C for 15 h. The suspension was centrifuged and washed with deionized water (200 mL) twice and once with ethanol (70 %). The white powder was dried (80 °C, ≥ 1 h) before it was ground in methanol (99 %) for 3 min. After drying again (80 ˚C, ≥ 1 h), the particles were calcined at 550 ˚C for 12 h.

### 2.1.3 Particle functionalization

In order to investigate the impact of contact angle on the ability of porous particles to nucleate ice via PCF, particles of similar pore diameters were functionalized with trimethyl and hydroxyl groups after calcination. We will focus on ice nucleation experiments with particles functionalized with trimethyl and hydroxyl groups rather than just calcined ones, because these were observed to change contact angle with ageing in air (Muster et al., 2001). A batch of 2.8 nm pore samples was calcined at 550 ˚C and then separated into three parts with one part unmodified, one part hydroxylated and the remaining part methylated. A summary of the particles investigated in this study is provided in Table 1. Hydroxylation and methylation were conducted as follows:

*Silanol surface (hydroxylation)*: A calcined sample (1.0 g) was suspended in toluene (200 mL) and heated to 60 °C before a calculated amount of water was added (Eq. A1 of Appendix A1) in order to achieve a concentration of 4.6 silanol groups nm$^{-2}$



(Zhuravlev, 2000). The particles were then suspended for 60 minutes through vigorous stirring, and occasional sonication, before they were filtered off, washed with deionized water (80 mL) and dried (120 °C, 20 mbar) overnight.

*Alkyl surface (trimethylation)*: A calcined sample (1.0 g) was suspended in toluene (200 mL at 60 ˚C before a 2-fold excess of organosilane (trimethylchlorosilane, 99 %, Sigma-Aldrich) as calculated using Eq. A2 (Appendix A1) was added. The reaction was run for 3 h and then the suspension was filtered and washed with toluene (50 mL), ethanol (50 mL), and water (50 mL). The particles were then dried (120 °C, 20 mbar) overnight.

## 2.2 Particle characterization

### 2.2.1 Nitrogen adsorption and calculation of pore size distribution

Particle surface area ($S_{BET}$) and pore diameters were determined by nitrogen adsorption (Quantachrome, NOVA 3000e). The nitrogen isotherms were obtained by measuring > 10 m² of dried (80 ˚C) sample and the $S_{BET}$ was obtained from the relative pressure range where multilayer adsorption takes place (0.05-0.30) and applying the BET equation (Brunauer et al., 1938). The average pore diameter ($d_{DFT}$) was obtained using the NLDFT (non-local density functional theory) method (Landers et al., 2013) applied to the N₂ sorption measurements.

### 2.2.2 DRIFTS

Diffuse reflectance infrared Fourier transform spectroscopy (DRIFTS) was used to characterise the functionalised particles and estimate the concentration of hydroxyl and methyl groups on the silica particles. The samples were prepared by combining 6 mg of dried sample (80 ˚C, > 1 h) with 194 mg of dry potassium bromide (KBr) to produce a 3 % (w/w) mixture. The mixture was ground vigorously for over a minute (Hamadeh et al., 1984) before being filled in the sample holder, where the sample was flattened with a spatula. A scan resolution of 4 cm⁻¹ was chosen and background scans with pure KBr were performed and each sample was corrected accordingly. The mixtures were scanned immediately after grinding to avoid the adsorption of water vapour. The background corrected scans were averaged and then normalized to the BET surface area of the sample instead of using the traditional method of normalizing based on the Si-O asymmetric stretching peak in the vicinity of 1100 cm⁻¹ (Muster et al., 2001). Normalization to the BET surface is more appropriate considering the porous nature of the samples.

### 2.2.3 Water sorption and contact angle derivation

Water sorption isotherms were obtained using dynamic vapour sorption (DVS, TA Instruments, VTI-SA+), where the water uptake is determined gravimetrically. Each isotherm was obtained using approximately 10 mg of sample dried at 120 ˚C in a pure nitrogen atmosphere for 1 h before the reference mass was determined in order to evaporate any pre-adsorbed water. The DVS cell was then cooled to the temperature ($T$ = 25 ˚C), at which the sorption measurements were performed. The adsorption isotherms were obtained by continuously measuring the sample mass while increasing the humidity from 0 to 90 % in steps of



5 % $RH_w$. The water uptake reported here denotes quasi equilibrium values at each $RH_w$ step defined as a mass change rate less than 0.008 % over the course of 5 min. The contact angle of the sample surface was then determined from the sorption isotherm using the Cohan-Kelvin equation (Kocherbitov and Alfredsson, 2007):

$$r_{nldft} - t_{ads} = -\frac{2\gamma(T)\cos(\theta)v_l(T)}{RTln(p/p_0)}. \tag{4}$$

5    Here $t_{ads}$ is the statistical thickness of adsorbed water, $r_{nldft}$ is the pore radius as determined by NLDFT ($d_{DFT}/2$) and $p/p_0$ is the water saturation ratio. The statistical thickness in cylindrical pores is calculated by subtracting the volume of the adsorbate ($V_{ads}$) from the full pore volume ($V_{tot}$) and can be rewritten as:

$$r_{nldft} - t_{ads} = \frac{r_{nldft}}{\sqrt{\frac{V_{tot}}{V_{tot}-V_{ads}}}}, \tag{5}$$

By substituting Eq. 5 into Eq. 4, the Cohan-Kelvin equation for cylindrical pores can be written as:

10    $$r_{nldft} = -\sqrt{\frac{V_{tot}}{V_{tot}-V_{ads}}} \cdot \frac{2\gamma(T)\cos(\theta)v_l(T)}{RTln(p/p_0)} \tag{6}$$

And when solving for $\theta$ becomes:

$$\theta = \arccos\left(\frac{r_{nldft}(RTln(p/p_0))}{-\sqrt{\frac{V_{tot}}{V_{tot}-V_{ads}}}2\gamma(T)v_l(T)}\right)\frac{\pi}{180}, \tag{7}$$

At 25 ˚C, $\gamma(T)$ = 71.69 mN/m and $v_l(T)$ is 20.5 m³/mol (Kocherbitov and Alfredsson, 2007), but differ from the values in bulk water due to being in confinement. $p/p_0$ is determined as the relative pressure where the pore condensation step is the 15 steepest.

## 2.3 Ice nucleation measurements

The mesoporous silica particles listed in Table 1 were tested in the Zurich Ice Nucleation Chamber (ZINC), a continuous flow diffusion chamber with a parallel plate geometry. The operating principal of ZINC can be found in Stetzer et al. (2008) and a brief description is given here. Aerosol particles are injected into ZINC where they become sheathed between particle-free 20 nitrogen in a region between two thermally controlled ice-coated walls. By applying a gradient in temperature between the two ice-coated walls, the temperature and supersaturation that the aerosols are exposed to is controlled. Depending on the aerosol properties and the set conditions in ZINC, aerosol particles may nucleate ice and continue to grow as they flow through the chamber until they reach an optical particle counter (OPC; Lighthouse Remote 3104) at the outlet of the chamber that counts and sizes the particles. All particles larger than 1 µm are considered ice crystals and are thus counted as ice nucleating 25 particles at the set conditions in ZINC. To ensure that the particles counted by the OPC are truly ice crystals and not water droplets when conditions exceed water saturation, the particles pass through an isothermal section kept at ice saturation (water subsaturation) and the temperature equivalent to the warm wall prior to being sampled by the OPC, allowing any formed cloud droplets to evaporate while the ice crystals remain unchanged.




All *RH*-scans between ice saturation and 105 % $RH_w$ were performed with a ramp rate of 2% increase in ice supersaturation per minute. At the start and end point of each scan, a 5-minute background sample was taken by forcing the sample flow through a filter in order to determine the background noise of the chamber. The OPC counts from these background periods were averaged and then linearly interpolated to produce a background that was subtracted from each *RH*-scan (Boose et al.,

2016; Burkert-Kohn et al., 2017). An activated fraction (*AF*) is calculated by comparing the number of particles larger than 1 μm exiting ZINC as determined by the OPC ($N_{ice(OPC)}$) and the number of aerosols entering the chamber ($N_{aero(CPC)}$), as counted by a condensation particle counter (CPC, TSI 3787) upstream of ZINC given by:

$$AF = \frac{N_{ice(OPC)}}{N_{aero(CPC)}}. \tag{8}$$

**2.4 Aerosol generation**

The particles were aerosolized using a rotating brush generator (Palas, RGB-100) and then passed through a 1 μm cyclone (URG-2000-30EHB) to further lower the chance of large particles proceeding through the system before entering a 2.7 m³ stainless steel tank (Kanji et al., 2013). The tank was filled to a concentration between 4000 and 10000 cm⁻³ and a fan inside the tank ensured that the particles remained suspended. Before entering ZINC the particles were size selected for 400 nm using a custom-built differential mobility analyzer (DMA), which consists of a polonium neutralizer and an electrostatic classifier

(TSI 3082 Long Column). Even though the synthesis procedure in this study produces a narrow particle size distribution, the DMA was used to remove any particles larger than 1 μm (from possible aggregation) to reduce the probability of misclassifying dry particles as ice crystals by the OPC.

**2.5 Differential scanning calorimetry**

In order to determine the ability of a critical ice embryo to fit in the pores of the samples tested, differential scanning

calorimetry (DSC; TA Instruments Q10) was performed. The DSC technique detects phase changes based on the heat flow associated with them (e.g., Kumar et al., 2018; Marcolli et al., 2007). Bulk samples were prepared by mixing between 1 and 5 mg of sample with ultrapure water (SigmaAldrich) or deionized water. Deionized water, which has a higher freezing temperature, was used for the large pore samples (9.1H2 and 9.0M2) to achieve a separation between the bulk water and pore water freezing peaks. All DSC experiments were conducted with a cooling rate of 5 K min⁻¹.

**3. Results**

The results are presented in three sections: the first characterizes the samples tested in this study (Sect. 3.1), the second investigates the ability of particles with 2.8 nm pores to nucleate ice depending on their contact angle (Sect. 3.2), and the third investigates the role of pore diameter on ice nucleation as a function of surface functionalization (Sect. 3.3).


## 3.1 Particle characterization

### 3.1.1 Particle surface area and pore diameter

Nitrogen adsorption and NLDFT provide particle surface area ($S_{BET}$) and average pore diameter ($d_{DFT}$), respectively, and are

5 summarized in Table 1 for each sample. The sample naming is such that the initial number represents the average pore diameter in nanometres followed by a C, M, or H to represent whether the sample was calcined, methylated, or hydroxylated, respectively. The numbers 1 or 2 after the letter indicate whether the samples are independent synthesis batches or several batches that have been combined and then separated and functionalized in different ways, respectively. An overview of the pore size distributions of the samples is shown in Fig. 1. As evident from Fig. 1a., the methylation of the 2.8 nm sample led to

10 a decrease in mean pore diameter by 0.1 nm (2.7M2). The presence of trimethylsilyl groups is confirmed by our DRIFTS measurements (see Sect. 3.1.2), indicating that the methylation was successful. However, we cannot quantify the exact coverage and distribution of the trimethylsilyl groups. The addition of hydroxyl groups to the silica does not produce a difference in the pore size relative to the calcined sample (Fig. 1a). This suggests that the OH groups do not detectably reduce the pore width or that the pore surface of the calcined sample is already sufficiently hydroxylated, as discussed below.

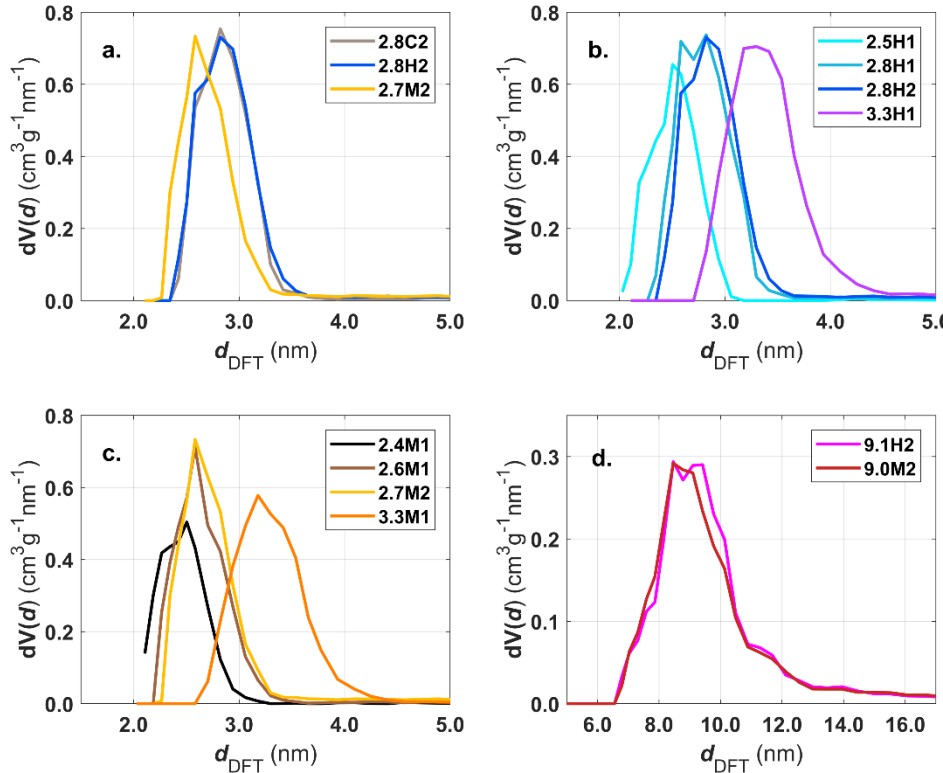

**Figure 1: Panel a shows pore size distributions for the 2.8 nm sample after calcination (grey), hydroxylation (blue) and methylation (gold). Panels b and c show pore size distributions of the hydroxylated and methylated samples, respectively. Panel d shows the pore size distribution for the SBA-15 samples after hydroxylation (magenta) and methylation (red).**





The pore size distributions of the hydroxylated samples are shown in Fig. 1b. The 2.8 nm samples, 2.8H1 and 2.8H2 are quite similar, however 2.8H1 has a larger fraction of 2.6 nm pores. 2.5H1 has the narrowest pore size distribution and the lowest total pore volume of the hydroxylated samples as shown in Table 1. Meanwhile, 3.3H1 has the broadest pore size distribution

with pores ranging from 2.7 to 4.5 nm. The methylated samples show a similar trend with a broadening of pore size distribution with increasing average pore diameter (see Fig. 1c). Consistent with functionalization, the methylated samples have lower pore volumes than the corresponding hydroxylated samples (see $V_{Ptot}$ in Table 1). This provides additional evidence that the methylation procedure was effective. The SBA samples (9.1H2 and 9.0M2) show a similar trend with a slight decrease in pore diameter upon methylation (Fig. 1d). As shown in Table 1, there is no relationship between the total BET surface area and

pore diameter except when comparing the MCM-41 to the SBA-15 samples which have approximately half of the specific surface area due to their differing morphology and pore structure.

**Table 1: Summary of samples used for ice nucleation studies. The BET method was used for total surface area ($S_{BET}$) and $\alpha_s$-plot for external surface area ($S_{EXT}$), (Bhambhani et al., 1972). The total pore volume ($V_{Ptot}$) was taken at $p/p_0$=0.95.**

| Sample Name | Synthesis Method | $d_{DFT}$ | Functionalization | $S_{BET}$ (m²/g) | $S_{EXT}$ (m²/g) | $V_{Ptot}$ (cm³/g) | $\theta$ (˚) |
|---|---|---|---|---|---|---|---|
| 3.3M1 | MCM-41 | 3.3 nm (±0.3) | Methyl | 726 | 12 | 0.50 | 75-80 |
| 3.3H1 | MCM-41 | 3.3 nm (±0.3) | Hydroxyl | 893 | 22 | 0.66 | 41-45 |
| 2.4M1 | MCM-41 | 2.4 nm (±0.2) | Methyl | 822 | 7 | 0.33 | 75-80 |
| 2.5H1 | MCM-41 | 2.5 nm (±0.2) | Hydroxyl | 892 | 7 | 0.38 | 41-45 |
| 2.6M1 | MCM-41 | 2.6 nm (±0.2) | Methyl | 917 | 12 | 0.42 | N/A |
| 2.8H1 | MCM-41 | 2.8 nm (±0.2) | Hydroxyl | 1007 | 15 | 0.53 | N/A |
| 2.7M2 | MCM-41 | 2.7 nm (±0.2) | Methyl | 925 | 13 | 0.45 | N/A |
| 2.8C2 | MCM-41 | 2.8 nm (±0.2) | Calcined | 868 | 12 | 0.49 | N/A |
| 2.8H2 | MCM-41 | 2.8 nm (±0.2) | Hydroxyl | 920 | 14 | 0.53 | N/A |
| 9.0M2 | SBA-15 | 9.0 nm (±1.1) | Methyl | 399 | N/A | 0.95 | 60-71 |
| 9.1H2 | SBA-15 | 9.1 nm (±1.1) | Hydroxyl | 429 | N/A | 0.98 | 15-37 |

### 3.1.2 DRIFTS

When comparing the impact of functionalization on the same initial bulk sample (2.8C2), the difference between hydroxylation (2.8H2) and methylation (2.7M2) is visible in the DRIFTS spectra (Fig. 2a). The intensity in the O-H stretching region, 3200-3800 cm⁻¹, is much larger for the hydroxylated sample (2.8H2) than the calcined (2.8C2) and methylated (2.7M2) samples, consistent with the addition of hydroxyl groups during the hydroxylation process. The broad absorption band peaking at about

3450 cm⁻¹ (3000-3700 cm⁻¹) in the calcined, hydroxylated and methylated samples is indicative of water adsorbed on the silica





surface and residual silanol groups (Chen et al., 1996). Previous studies have shown that calcining silica particles at temperatures above 200 ˚C, as is the case for our calcined samples, removes all free water (Muster et al., 2001; Zhuravlev, 2000). However, here the DRIFTS cell was operated at ambient conditions, allowing for water to (re-)adsorb to the particle surface and contributing to the broad absorption in the range 3000 – 3700 cm$^{-1}$ (Muster et al., 2001). Indeed, when exposing

a silica sample calcined at 200 °C to ambient conditions, the increase in mass due to adsorbed water is visible using thermogravimetric analysis (not shown). The methylated sample (2.7M2) has the weakest absorbance in the OH stretching region (3000-3700 cm$^{-1}$; see Fig. 2a). Furthermore, the methylated sample shows a peak associated with the C-H stretching band around 2960 cm$^{-1}$, indicating the presence of trimethylsilyl groups bonded to the silica surface. However, the presence of isolated and geminal silanol groups, as shown by the peak at 3750 cm$^{-1}$, indicates that the methylation is incomplete (Bergna,

1994; Muster et al., 2001). The increase in the C-H stretching band due to the methylation (at 2960 cm$^{-1}$), roughly corresponds to the decrease in the isolated silanol/geminal silanol peak for the 2.8 nm samples (Fig. 2a). The calcined sample (2.8C2) has the highest concentration of isolated and geminal silanol groups. This is expected as during hydroxylation (2.8C2 transitioning to 2.8H2) the concentration of silanol groups increases and becomes sufficiently dense for chains of hydrogen bonds to form between individual silanol groups, decreasing the number of isolated silanol groups and thereby shifting the peak at 3750 cm$^{-}$

$^1$ to ~3660 cm$^{-1}$ (Muster et al., 2001).



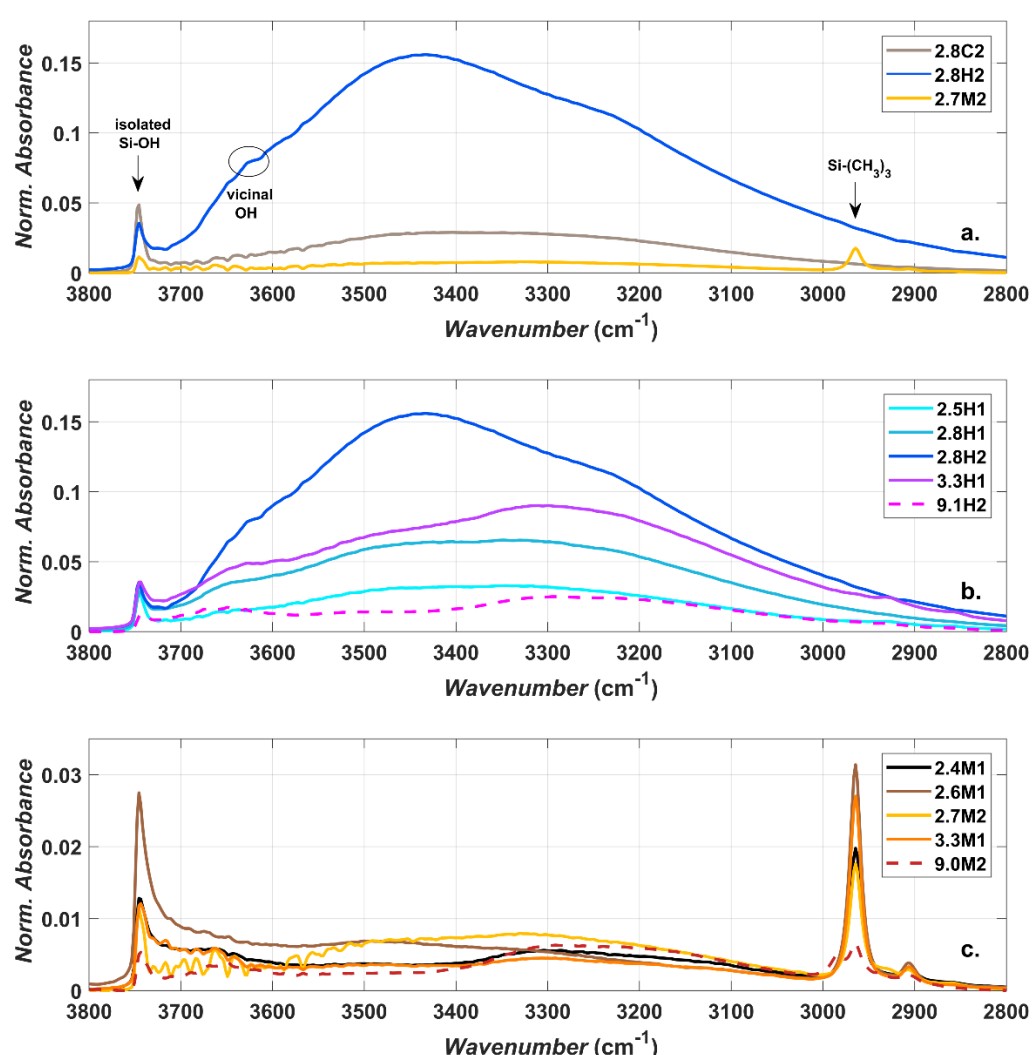

**Figure 2: DRIFTS normalized absorbance (Kubelka-Munk) for the 2.8 nm particles after calcination (grey), hydroxylation (blue) and methylation (gold) (panel a). Panels b and c show the spectra of hydroxylated and methylated samples, respectively.**

When comparing the DRIFTS results of the different hydroxylated and methylated samples (Fig. 2b and c), it is clear that the

5   SBA-15 particles (9.1H2 and 9.0M2) also show absorbance in the OH (3200-3800 cm$^{-1}$) and CH (~2960 cm$^{-1}$) stretching region

of the spectra, respectively, demonstrating that the functionalization was successful. Differing peak intensities between particle

types could be due to the differing densities of silanol and siloxanes on the surface of the particles. Although the clear peak in

the C-H stretch region of the DRIFTS (Fig. 2c) shows that the methylation process on the SBA-15 particles (9.1H2

functionalization to 9.0M2) was successful, methylation is far from complete since the peak arising from isolated/geminal

10   silanol stretching vibrations (3750 cm$^{-1}$) is still visible in all methylated samples (Fig. 2c). The concentration of hydroxyl

groups on the hydroxylated samples is independent of pore size (Fig. 2).  Rather, the intensity in the O–H stretching region

likely depends on the age and exposure of the calcined samples to ambient water vapour. The methylated samples show much




less spread in the amount of adsorbed water, suggesting that they are more resistant to hydroxylation and more stable over time (Fig. 2c).

### 3.1.3 Water vapour sorption

Water vapour sorption isotherms obtained for the samples 2.4M1, 2.5H1, 3.3M1, 3.3H1, 9.1H2 and 9.0M2 are shown in Fig. 3. The sorption isotherms have been classified following the recommendation by the International Union of Pure and Applied Chemistry (IUPAC, (Sing, 2009; Thommes et al., 2015). The hydroxylated samples (2.5H1, 3.3H1 and 9.1H2) show Type IV isotherms, characterized by an initial monolayer-multilayer adsorption occurring on the pore wall followed by a steep, almost step-like increase in water mass known as the condensation step at the $p/p_0$ or $RH$ associated with pore filling. This is consistent with previous observations for mesoporous silica (Kittaka et al., 2011). The methylated samples, 2.4M1, 3.3M1 and 9.0M2 show similar isotherms, but lack an initial monolayer adsorption along the uptake curves. This is consistent with a Type V isotherm and provides direct evidence that the methylation was successful in making the particles more hydrophobic. It should be highlighted that in the 2$^{nd}$ sorption cycle, the methylated samples have Type IV isotherms that are more similar to the isotherms of the hydroxylated samples, independent of the pore size. This transition suggests that the exposure to high concentrations of water vapour during the first sorption cycle increases the number of silanol groups on the surface of the methylated samples. Indeed, the second sorption cycle of the methylated samples shows that the condensation step shifts close to the $RH$ of the hydroxylated sample during the first sorption cycle. This indicates that the contact angle of the methylated sample becomes closer to that of the hydroxylated sample.

Similarly, the shift in the condensation step to lower humidities for the hydroxylated samples suggests a decrease in contact angle. The relative mass of the hydroxylated samples do not return to zero after the desorption cycle (Fig. 3a and b), indicating that water remains adsorbed on the particles. This strongly adsorbed water is expected to lower the contact angle between water and the wall surface to nearly zero. Moreover, multilayers of adsorbed water narrow the effective diameter for pore filling (Broekhoff and de Boer, 1967; Kruk et al., 1997; Miyahara et al., 2000). Both effects explain the observed shift of the condensation step to lower humidities. Furthermore, it is visible from Fig. 3 that the hydroxylated samples adsorb relatively more water than the methylated samples even though they have very similar pore diameters (see $d_{DFT}$ in Table 1). However, the samples have differing total pore volumes ($V_{Ptot}$) and thus it is expected that the absolute amount of condensed water differs.



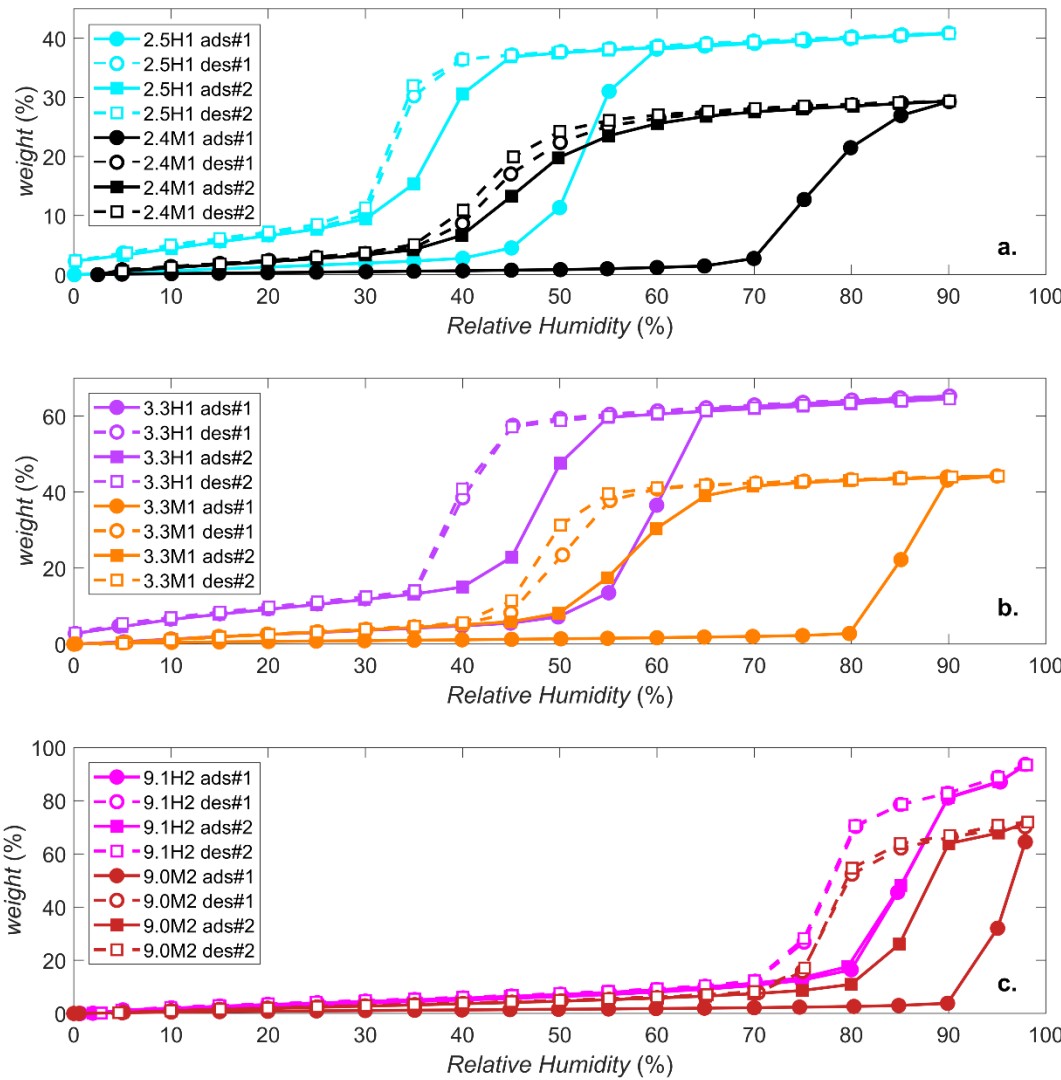

**Figure 3: Water sorption isotherms for 2.5H1 (cyan) and 2.4M1 (black) in panel a., 3.3H1 (purple or lilac) and 3.3M1 (orange) in panel b., and 9.1H2 (magenta) and 9.0M2 (dark red) in panel c. The solid and dashed lines with closed and open symbols indicate adsorption and desorption isotherms, respectively. The first and second adsorption/desorption cycles are indicated by circles and squares, respectively.**

The contact angle of the samples is obtained by inserting the *RH* of the condensation step in the first water sorption cycle into Eq. 7 (see Table 1). The contact angles for the MCM-41 particles ranged between 41˚- 45˚ and 75 ˚- 80˚, for the hydroxylated and methylated samples, respectively. Conversely, the SBA-15 type samples have significantly lower contact angles of 15˚ and 60˚ for the hydroxylated (9.1H2) and methylated (9.0M2) samples, respectively. However, these values may be low-biased due to the spread of pore diameters within the sample. As can be seen from Fig. 1, the pore size distribution is significantly wider for the SBA-15 samples than for the MCM-41 particles (ranges from ~7 to 9 nm at the max). Therefore, it is difficult to properly assign the correct pore diameter responsible for the initial pore condensation observed from the sorption

### 3.1.4 DSC measurements

Upon cooling of a sample prepared as a slurry in the DSC, the exterior water freezes first followed by the freezing of pore water due to the decrease in temperature required for water in confinement to freeze (Deschamps et al., 2010; Janssen et al., 2004; Jelassi et al., 2010; Kittaka et al., 2011; Marcolli, 2014; Moore et al., 2012; Morishige and Uematsu, 2005). This can most clearly be seen in the freezing of 3.3M1 shown in Fig.4a where the initial release of latent heat (peak) centred around 255 K is due to the freezing of the exterior bulk water followed by the second peak starting at 234 K due to the freezing of the

pore water. Tap water was used for the experiments with the SBA-15 samples to shift the freezing of exterior water to higher temperatures so that the freezing of pore water is observable (Fig 4c). Also shown in Fig. 4 are the expected critical pore diameters $D_p$ calculated following Eqs. 2 and 3, yielding a direct comparison of the theoretical predictions of pore freezing to the experimentally determined onset of ice formation (peaks in thermograms). Since previous MD simulations and X-ray diffraction studies have shown ice in confinement to be typically stacking disordered or cubic (Moore et al., 2010, 2012;

Morishige et al., 2009) two parameterizations from literature (Murray et al., 2010; Zobrist et al., 2007) were used to calculate $D_p$, assuming either cubic (Fig. 4a and b) or hexagonal (Fig. 4c). As can be seen in Fig. 4, the parameterization assuming cubic ice is more accurate at predicting the observed freezing temperature for narrow mesopores (2.8 and 3.3 nm samples) where the freezing temperatures are around 230 K. Whereas the freezing temperature of the 9.1 nm pore samples (9.1H2 and 9.0M2; Fig 4c) of approximately 261 K is better predicted assuming that the ice is hexagonal. These results are consistent with studies that

have shown that cubic ice occurs more readily at colder temperatures (Kuhs et al., 2012; L. Malkin et al., 2015).

The DSC thermograms of the SBA-15 samples (9.1H2 and 9.0M2) show a bimodal peak associated with the freezing of pore water (see Fig. 4.c), with a pronounced peak around 258 K and a shoulder towards higher $T$. This indicates that there is a bimodal distribution of pore sizes that contribute differing fractions of pore volume to the samples. Indeed, the pore size

distributions show that the SBA-15 samples have a clear shoulder in the distribution at 11 nm followed by a main peak at 9.1 nm (see Fig. 1.d). Thus the bimodal freezing signal is likely due to the freezing of pore water in pores larger than 11.0 nm followed by the release of heat from the freezing in the smaller, more abundant 9.1 nm, remaining pores.

Deschamps et al., (2010) showed that highly hydrophobic pore surfaces had lower melting and freezing temperatures than

hydrophilic pores of the same diameter. They associated the depressed freezing temperatures with a decrease in mobility of water molecules in hydrophobic mesopores. However, as the pore size exceeded 3 nm, the dependence on pore-hydrophobicity of the freezing point depression disappeared (Deschamps et al., 2010; Jähnert et al., 2008; Schreiber et al., 2001). In agreement with a loss of the dependence on hydrophobicity for pores larger than 3 nm, Moore et al, (2012) showed that the melting



temperature in a 4 nm diameter silica pore was the same regardless of hydrophobicity using MD simulations. The DSC thermographs in Fig. 4 show that there is no detectable difference in the onset freezing temperatures for the SBA-15 samples depending on functionalization (9.1H2 and 9.0M2; Fig. 4c). However, in contrast to the results of Deschamps et al. (2010), the 2.8 nm (Fig. 4b) samples also show no detectable difference in freezing onset. This indifference may stem from the fact

5  that the observed freezing onsets occur due to the ice growth in the largest detectable pores of a sample. As can be seen from Fig. 1, both 2.8H2 and 2.7M2 contain a fraction of pores larger than 3 nm. Therefore, the observed freezing onsets in the thermograms may be due to pore diameters that are wide enough for growing ice to not be impacted by the contact angle of the pore wall (Moore et al., 2012).

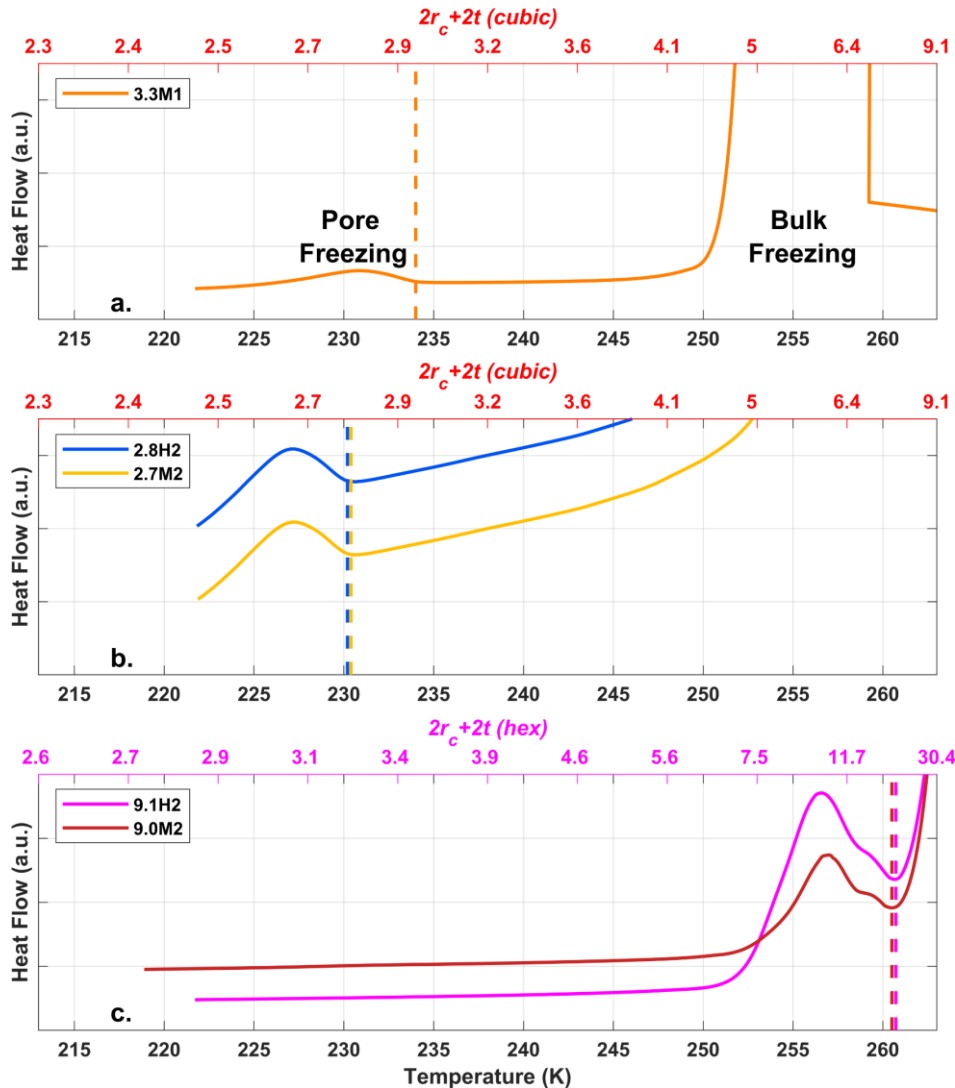

10  **Figure 4: DSC thermograms for ice growth into pores for samples 3.3M1 (panel a), 2.8H2 and 2.7M2 (panel b), and 9.1H2 and 9.0M2 (panel c). The vertical dashed lines mark the observed onset temperatures of pore freezing. The upper x-axes represent the predicted critical pore diameter *(2r$_c$+2t)* for cubic (red axis labels, panel a and b) or hexagonal (magenta axis labels, panel c) ice to be stable,**





**following Marcolli (2014). The peak on the right-hand side of panel a and the descending lines in panel c are due to the bulk freezing of exterior water.**

## 3.2 Pore condensation and freezing experiments: the 2.8 nm pore example

A summary of the ice formation activity of the functionalized 2.8 nm particles from a single batch (2.8H2, 2.8C2 and 2.7M2) is shown in Fig. 5, where the $RH_i$ required for an $AF$ of 0.05 ($AF_{0.05}$) are shown. The complete $AF$ curves are shown in Appendix A1. The onset $RH_i$ at 223 and 228 K is significantly lower for the hydroxylated (blue) and calcined (grey) samples than for the methylated sample (gold). This reveals a strong dependence on the contact angle, which is lower in case of the hydroxylated sample ($\theta = 41° - 45°$) compared to the methylated sample ($\theta = 75° - 80°$). The lower onset humidity of the hydroxylated sample is consistent with the inverse Kelvin effect, which predicts pore filling to occur at a lower $RH_i$ for the hydroxylated sample. Furthermore, the pore filling line based on Eq.1 for the methylated sample (assuming $\theta = 78°$), predicts the observed freezing onsets at 223 and 228 K (gold line, $AF_{0.05}$), respectively, within experimental uncertainty, indicating that the PCF mechanism is limited by pore filling. Additionally, this suggests that the ice formed in the methylated pores investigated here is capable of growing into the unconfined vapour region as proposed by David et al. (2019) without the need for a two-step nucleation process (Christenson, 2013; Kovács and Christenson, 2012; Page and Sear, 2006). In contrast, the 2.8 nm hydroxylated sample (2.8H2) is predicted to fill below ice saturation ($RH_i \geq 71$ % at 223 K and $RH_i \geq 69$ % at 228 K). Therefore, ice growth should be observed as soon as ice saturation is exceeded within ZINC, yet a $RH_i$ of 118% (223 K) and 112 % (228 K) is required to observe an $AF_{0.05}$ (Fig. 5). This might suggest that a two-step nucleation mechanism is required for ice to grow out of narrow calcined and hydroxylated mesopores at these lower supersaturations (Campbell et al., 2017; Christenson, 2013). However, particles must grow to at least 1 μm before they are detected as ice crystals in this study (see Section 2.3). Hence, the limited growth time in ZINC (~10 seconds) for the particles to reach a size of 1 μm must be accounted for when interpreting ice onset. Therefore, we calculated theoretical ice growth curves (dashed peach lines, see A2 for calculation) using the parameterization from Rogers and Yau (1989) and assuming accommodation coefficients of 0.1 and 0.2 for ice growth at $T < $ HFT (Skrotzki et al., 2013). The ice crystal shape in the growth calculation was assumed to be spherical due to the small final size and its growth on spherical particles (Järvinen et al., 2016). Comparing our ice onsets to the expected growth (dashed peach lines, Fig. 5), it immediately becomes clear that the slow ice crystal growth explains the required $RH_i$ to observe ice within the ZINC experiments, without the need of a two-step nucleation mechanism (David et al., 2019).





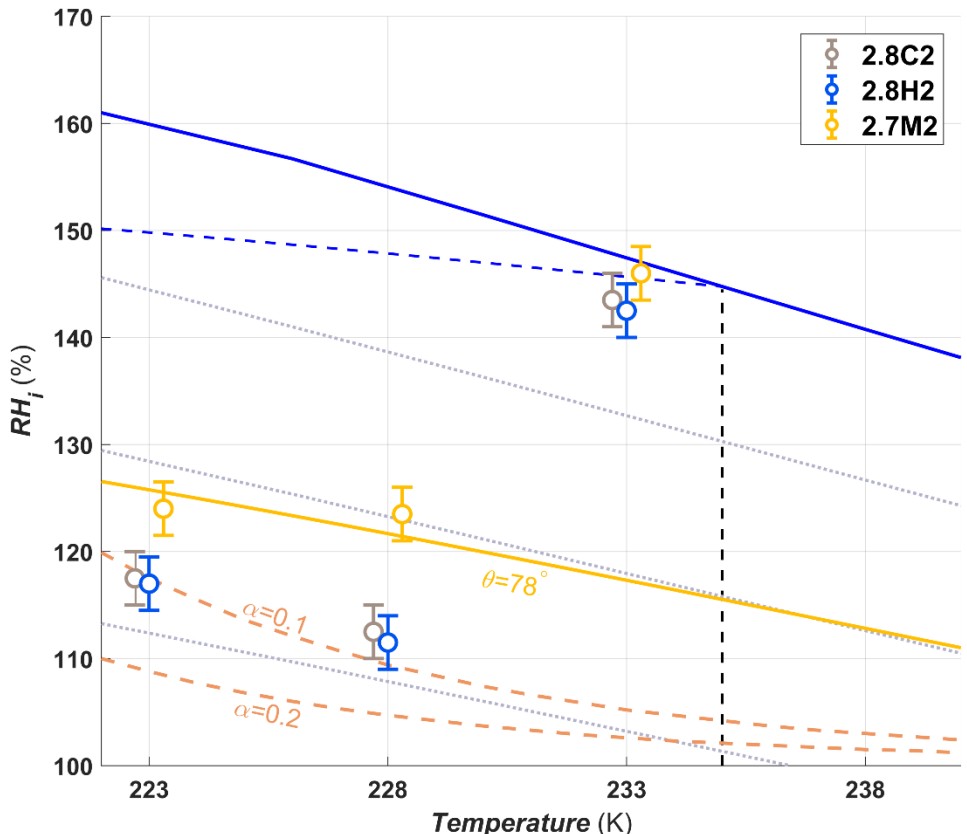

**Figure 5: The average $RH_i$ required for an $AF_{0.05}$ for the hydroxylated (blue), methylated (gold) and calcined (grey) porous (2.7 –**
**2.8 nm) silica samples. The error bars represent the $RH_i$ uncertainty in ZINC. The solid blue line and dotted grey lines represent**
**water saturation and constant $RH_w$ decreasing in steps of 10 % from the water saturation line respectively. The dashed blue line is**
**the homogeneous nucleation $RH_i$ based on Koop et al. (2000) assuming a nucleation rate of $10^8$ cm$^{-3}$ s$^{-1}$. The vertical dashed black**
**line represents the homogeneous freezing temperature of pure water (HFT). The golden line denotes pore filling of 2.7 nm wide**
**pores assuming a contact angle of $\theta = 78°$. Pores of the hydroxylated and calcined samples are expected to fill well below ice**
**saturation. The dashed salmon lines indicate the required $RH_i$ for ice to grow to a detectable size within the residence time of ZINC**
**assuming an accommodation coefficient ($\alpha$) of 0.1 or 0.2, respectively. Measurements were also conducted at 238 K but a $AF_{0.05}$ was**
10 **not detected. The symbols of the calcined and methylated samples are offset by 0.3 K colder and warmer, respectively, for clarity.**

In a PCF mechanism it is expected that once the critical humidity for pore filling is reached, ice nucleation and growth should

lead to a step-like increase of $AF$ to values close to unity. Yet the increase in $RH_i$ between onset and $AF_{0.05}$ (Figs. A1 and A2)

indicates that either the pores on the particles are not homogeneous in both pore size and/or contact angle or the conditions

within ZINC are non-uniform. Indeed, Garimella et al. (2017) showed that turbulence at the entrance of the chamber causes

particles to leave the (predicted) aerosol lamina and therefore, these particles experience a lower $RH_i$ than the ones in the

lamina. The assumption that all particles are within the lamina leads therefore to a low bias of $AF$ at all $RH_i$ and a shift of $AF=1$

to higher $RH_i$. The underestimation of $AF$ depends on the set temperature and ice supersaturation (gradient between the wall

temperatures). At 233 K and 102% $RH_w$ only 78% of the particles were shown to be exposed to the set conditions in ZINC

(Garimella et al., 2017). Particles deviating from the lamina likely explain the gradual increase of $AF$ with increasing $RH_i$.



The shift of $AF_{0.05}$ close to water saturation at 233 K (Fig. 5) is consistent with the DSC measurements which show that the pore water only starts to freeze at 230 K for 2.7 – 2.8 nm pore diameter samples (Fig. 4b), suggesting that the pores are too narrow for PCF to occur at 233 K. Rather bulk water on the particle surface is required for homogeneous freezing consistent with the observed shift of $AF_{0.05}$ to water saturation at 233 K.

When examining the entire $AF$ curves at 233 K shown in Figs. A1c and A2c, there is a clear increase in $AF$ to approximately 0.02 for the hydroxylated sample (2.8H2) between 100 and 120% $RH_i$, and a slight increase of 0.002 for the methylated sample (2.7M2). This suggests that a fraction of the pores small enough to remain undetected in the DSC (see Section 3.1.4) nucleate ice homogeneously at 233 K because they exhibit diameters > 3 nm which is large enough to accommodate the critical ice germs. Such an assumption is in agreement with the tail of pores with $d_{DFT} > 3$ nm appearing in the pore size distribution of the samples in Fig. 1.a. Following Marcolli (2014) for calculating $D_p$ (see Eq. 3), cubic or stacking disordered ice should be stable in pores of approximately 3 nm diameter at 233 K (see Fig. 4.b; Moore et al., 2010; Morishige et al., 2009). However, even if the pore diameter is large enough to host a critical ice germ, ice may fail to form during the residence time in ZINC when the nucleation rate within the pore water is too low. Using rates for homogenous ice nucleation derived from experiments ($J_{hom}(T)$; Atkinson et al., 2016; Ickes et al., 2015; Riechers et al., 2013), the residence time in ZINC ($t_{ZINC}$) for a given pore volume ($V_{pore}$) to nucleate ice can be calculated as:

$$t_{ZINC} = \frac{-\ln(1-AF)}{J_{hom}(T)V_{pore}}. \tag{9}$$

Using $J_{hom}(T)$ of $10^{10\text{-}12}$ cm$^{-3}$s$^{-1}$ at 233 K as reported in literature (Ickes et al., 2015; Koop and Murray, 2016; Murray et al., 2010), and $V_{pore}$ based on a single pore with average width and the length of an average particle (400 nm), the residence time in ZINC would need to be 3 to 4 orders of magnitude longer than the ~ 10 s available in order to reach $AF_{0.05}$. Therefore, the observed increase in $AF$ to 0.02 for 2.8H2 (Fig. A1c) does not comply with reported homogeneous ice nucleation rates, but may be explained by the effect of pressure. As RH decreases, tension (negative pressure) builds up in the pore water as a function of the curvature of the water meniscus at the pore opening, such that nucleation rates increase drastically (Marcolli, 2019). At the $RH$ of pore filling ($RH_w = 67$ %), the pore water experiences a strongly negative pressure (-83 MPa). At 233 K and water saturation ($P = 0.1$ MPa), the pressure-dependent extension of the Murray et al. (2010) parameterization of CNT (Marcolli, 2019) predicts a nucleation rate of $9 \cdot 10^{10}$ cm$^{-3}$s$^{-1}$ (when using an exponent of n = 0.97 for estimating $\sigma_{iw}$ following Murray et al., (2010)). However at $RH_w = 67$ % ($P = -83$ MPa) the nucleation rate increases to over $10^{21}$ cm$^{-3}$s$^{-1}$ ($4 \cdot 10^{22}$ cm$^{-3}$s$^{-1}$), which should result in freezing of pore water in less than a millisecond. Note that the dry particles that are injected into ZINC reach ice saturation condition within the chamber after about 0.5 s and water saturation condition after about 1 s (Stetzer et al., 2008). Thus, there should be enough time for pore water to freeze before equilibrium conditions are reached for those pores that are wide enough to host ice.





At 238 K, the 2.8 nm samples discussed in this section (2.8H2, 2.8C2 and 2.7M2) do not reach $AF_{0.05}$ even above water saturation (Fig. A1.d and A2.d). Only as the $RH_i$ approaches and exceeds water saturation the $AF$s of 2.8H2 and 2.7M2 reach 0.04 and 0.006, respectively (Figs A1d and A2d). The curves of the hydroxylated samples show a weak increase that steepens when water saturation is approached (Fig. A1d), which we ascribe to condensation or immersion freezing (Vali et al., 2015)

occurring at active sites on the external particle surface. Pore water is likely not responsible for the observed freezing when the entire particle is immersed in water (Campbell et al., 2015). Furthermore, as temperature increases, the critical ice germ size increases and therefore, a larger pore diameter is required for PCF to occur at 238 K than at 233 K independent of the presence of active sites in the pore. Based on the DSC thermograms shown in Fig. 4a, even the samples with pore diameters of 3.3 nm have freezing onsets below 238 K. Moreover, 9.1H2 with pore diameters large enough to accommodate the critical

ice embryo based on the DSC measurements (see Fig. 4c) is least efficient at freezing (see Fig. A1d). Therefore, the observed freezing is likely due to nucleation sites on the outer particle surface that become active when immersed in water. This assumption is further substantiated by experiments performed with nonporous silica particles which showed similar ice nucleation activity as their porous counterparts (not shown) at 238 K. The ability of these active sites to nucleate ice is reduced on the methylated sample (see Fig. A2), consistent with previous studies that suggest that hydroxyl groups are important for

templating ice formation (Pedevilla et al., 2017) and alkylated silica surfaces suppress ice nucleation (Kanji et al., 2008).

### 3.3 Role of pore diameter on PCF

In order to investigate the ability of the pore diameter to influence the PCF mechanism, particles with different pore diameters were synthesized with either hydroxyl or trimethylsilyl surface groups and we discuss these separately in the following.

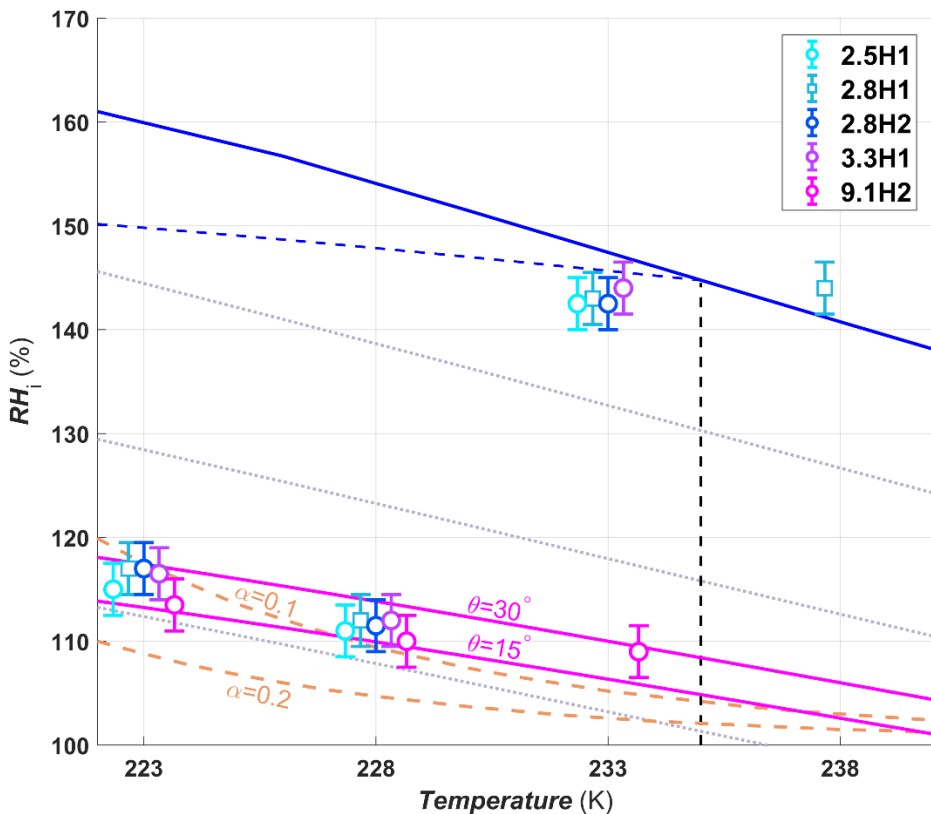

**Figure 6: Average $RH_i$ required to reach an $AF_{0.05}$ conditions for hydroxylated silica samples of different pore diameters. The symbols are offset by ± 0.3 or 0.6 K from the experimental temperatures to make the points more visible. The pore filling lines are plotted for 9.1 nm pores assuming a contact angle of 15 and 30˚ (magenta lines). Symbols and reference lines are as in Fig. 5.**

**3.3.1 Hydroxylated samples**

Pore diameter has no impact on the humidity required for the hydroxylated samples to reach $AF_{0.05}$ below the HFT (Fig. 6, dashed back line), the only exception being at 233 K, where ice formation starts well below water saturation for the 9.1H2 sample, while the smaller pore size samples reach $AF_{0.05}$ only close to water saturation. This is indeed expected for pores up to 3.3 nm, when considering that the contact angle of the pore surface is rather low for the hydroxylated samples (41 - 45˚; see

Sect. 3.1.3), such that the pores are expected to fill already below ice saturation. Assuming that 9.1H2 has a similar contact angle as the other hydroxylated samples, the 9.1 nm pores would require a $RH_i$ of ~123 and 118 % at 223 and 228 K, respectively, for pore filling to occur. However, based on the water sorption measurements (see Fig. 3c) the estimated contact angle of 9.1H2 is approximately 15˚. Furthermore, when examining the pore size distribution shown in Fig. 1d, more than 5 % of the pores are smaller than 9.1 nm (between 7 and 9 nm) and thus a lower humidity for pore filling is required for these pores.

Based on the lower contact angle alone, the 9.1 nm pores are expected to fill at ~114 and 109 % $RH_i$ at 223 and 228 K, respectively. Therefore, no significant dependence of $AF_{0.05}$ is expected for the investigated samples at 223 and 228 K due to the ice growth limitations in ZINC, as discussed above.





At 233 K the $AF_{0.05}$ $RH_i$ shifts to water saturation for all samples except for 9.1H2 (Fig. 6). The DSC experiments with 3.3M1 show that ice freezes within the pores only below 233 K (Fig. 4a). Therefore the inability for 5% of the particles to freeze up to water saturation for these samples, is consistent with PCF.

5 In contrast, the pore diameters in the 9.1H2 sample are wide enough to host ice at 233 K (see DSC experiments, Fig. 4c). Yet, nucleation rates at this temperature are too low for pore water to freeze within the residence time of ZINC. However, the strong increase of $AF$ at about 108 % $RH_i$ together with the decrease starting from 125 % $RH_i$ (see Fig. A1c) can be explained by the dependence of nucleation rates on pressure (Marcolli, 2019). At 233 K, the pressure dependent version of the Murray et al., (2010) parameterization of CNT (Marcolli, 2019) predicts a nucleation rate of $9 \cdot 10^{10}$ cm$^{-3}$s$^{-1}$ at water saturation ($P = 0.1$ MPa), 10 which increases to $2 \cdot 10^{16}$ cm$^{-3}$s$^{-1}$ at $RH_i$ of pore filling (~108 % $RH_i$ , $P = -32$ MPa). Thus, the water in a pore of 9.1 nm diameter should freeze at 108 % $RH_i$ within about 1.6 s. At water saturation, it takes one pore $4 \cdot 10^5$ s to freeze, implying that most pores should freeze at 108 % $RH_i$ within the residence time of ZINC and that $AF$ decreases when water saturation is approached as can be seen in Fig. A1c.

At 238 K only the 2.8H1 sample reached an $AF_{0.05}$ (Fig. 6). However, all of the other hydroxylated samples with the exception 15 of 9.1H2, have a similar increase in $AF$ near water saturation reaching values just below the $AF_{0.05}$ threshold (Fig. A1d). This indicates that there are active sites located on the external particle surface that nucleate ice through immersion or condensation freezing rather than PCF. Considering that pores are closely spaced, the outer surface is cladded in pore openings, providing a nanoscale pattern that might influence the ice nucleation activity in immersion and condensation mode.

20 The SBA-15 sample 9.1H2 also showed an increase in $AF$ near water saturation, albeit the increase was about an order of magnitude lower than for the MCM-41 samples (Fig. A1d). Thus, the 9.1H2 surface seems to be less efficient at nucleating ice than the one of the MCM-41 samples, indicating that the synthesis procedure for SBA-15 particles (see Sections 2.1.1 and 2.1.2), generates less active sites than the one for MCM-41 samples. This is especially true when considering that the pores in the MCM-41 samples are too narrow to host the critical ice embryo and therefore, the surface area is significantly smaller than 25 for the 9.1H2 sample. In experiments performed at 243 K the ability of the hydroxylated samples to nucleate ice approached the detection limit of ZINC and are therefore not show.

### 3.3.2 Methylated samples

Unlike the hydroxylated samples, the methylated samples show a dependence of onset humidity on pore diameter. At 223 and 228 K, the samples with 2.4 nm pore diameters had the lowest $AF_{0.05}$ $RH_i$ followed by the 2.6, 2.7 and 9.0 nm samples, which 30 overlap, and lastly the 3.3 nm sample (Fig. 7). This indicates that due to the higher contact angles after methylation (60˚ – 71˚ compared with 15˚ – 37˚ for the hydroxylated samples), PCF is limited by pore filling rather than ice growth to the detection limit of ZINC. Thus, the increase in $RH_i$ required for filling of increasing pore diameters, is observable within ZINC for methylated samples. The similar $AF_{0.05}$ $RH_i$ of 9.0M2 at 223 and 228 K compared with the methylated MCM-41 samples can





be explained by its lower contact angle (60˚ vs. 78˚, see Table 1) suggesting a more hydrophilic surface of 9.0M2 compared with MCM-41. The 9.0M2 is less active than the 9.1H2 sample (see Figs. 6 and 7), indicating that the methylation and subsequent increase in contact angle decreased the ice nucleation ability.

Unlike the hydroxylated samples at 233 K, which showed a clear increase in $AF$ at $\sim$ 105 % $RH_i$ (Fig. A1c), the methylated samples (except 9.0M2) show only a weak and continuous increase in $AF$ up to approximately 0.01 before water saturation is reached and homogeneous freezing of bulk water sets in (see Fig. A2c). This difference is likely due to the higher humidity required for pore filling of the methylated samples. The pore water experiences just a moderately negative pressure of at most -26 MPa at pore filling conditions, which enhances nucleation rates to a level that is able to induce freezing in only very few
pores that also need to be wide enough to host ice. The reduction in $AF$ below water saturation of the methylated compared with the hydroxylated samples is consistent with previous observations that alkylation of silanol groups suppressed ice nucleation below water saturation at 233 K (Kanji et al., 2008).

The $AF_{0.05}$ of 9.0M2 is close to the predicted pore filling line for $\theta = 78$˚ at 233 K (Fig. 7), while at 223 and 228 K, it is much
below the predicted line even for $\theta = 60$˚. This freezing activity is attributed to either non-uniform methylation which led to variations in contact angle or the presence of pore-like imperfections on the rough surface of the 9.0M2 particles that are narrower than the measured pore diameters and remained undetected in the pore-size distribution (Fig. 1) due to their extremely small volumes. At 233 K, the onset humidity of the 2.7M2 shifts close to water saturation in accordance with the DSC results in Fig. 4 showing that ice only freezes close to or below 233 K for pores narrower than 3.3 nm (see Fig. A2). A clear onset
$RH_i$ where the bulk of the 9.0M2 particles nucleate ice is absent (Fig. A2c) as is observed with 9.1H2 in Fig. A1c, indicating that pores of 9.0M2 continuously fill and freeze while RH increases. Indeed, the measured pore diameters by $N_2$ sorption (see Fig. 1d) show a wide pore size distribution for 9.0M2 and 9.1H2. Water uptake reaching 1 wt% only above 90 % $RH_w$ for 9.0M2 (Fig. 3), is in accordance with $AF$ exceeding 0.05 only for $RH_i > 130$ % further supporting that ice nucleation on 9.0M2 is limited by pore filling. Since homogeneous nucleation rates close to water saturation at 233 K are rather too low to induce
freezing of water in 9.0M2 pores, ice nucleation sites for immersion freezing within the pores might be responsible for the observed $AF$. The assumption of nucleation sites on the 9.0M2 is further supported by its ice nucleation activity persisting above the HFT (Fig. A2d).



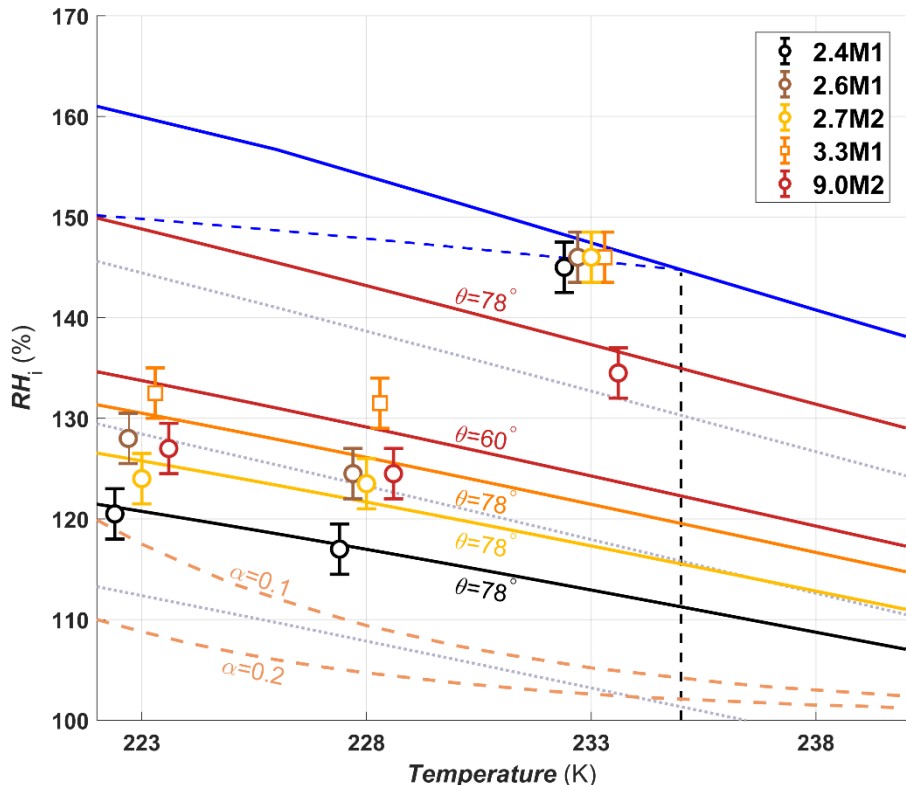

**Figure 7:** Ice formation onset conditions for methylated silica samples of different pore diameters. The symbols are offset by ± 0.3 or 0.6 K from the experimental temperatures to make the points more visible. Pore filling lines are given for 2.4 (black line), 2.7 (gold line), and 3.3 (orange line) nm pores assuming a contact angle of 78˚. For 9.0 nm pores the pore filling lines are plotted assuming contact angles of 60˚ and 78˚ (red lines).

The methylated MCM-41 samples have a very weak increase in $AF$ around water saturation at 238 K (Fig. A2d), reaching similar $AF$ as at 233 K below water saturation. The increase of $AF$ for MCM-41 samples at 238 K is lower than the one observed for the hydroxylated MCM-41 samples at the same temperatures and relative humidities, consistent with the notion that methylation decreases the ice nucleation activity of a surface (Kanji et al., 2008).

At 238 K, the 9.0M2 sample shows a distinct and continuous increase of $AF$ below water saturation. Since the temperature is too high for homogeneous ice nucleation within pores, this is a clear indication of active sites present in pores resulting in immersion freezing as soon as the pores fill with water. Interestingly, the $AF$ is higher in 9.0M2 compared with 9.1H2 (Fig. A2d and Fig. A1d, respectively), showing that the density of hydroxyl groups is not always a good predictor for ice nucleation ability. Indeed, it has been shown that methylated amorphous silica has an enhanced nucleation ability relative to hydroxylated silica due to the condensation of water on a hydrophilic Si-OH group surrounded by methylated groups (Bassett et al., 1970; Salazar and Sepúlveda, 1983). Salazar and Sepúlveda, (1983) postulated that adsorption on islands of silanol groups followed by multilayer growth similar to condensation of water would nucleate ice when water molecules come in contact with the neighbouring methyl groups. However, it is important to note that hydroxylation and methylation had an opposite effect on the heterogeneous ice nucleation ability at 238 K of MCM-41 and SBA-15 particles, making generalization in terms of





dependence on contact angle and degree of hydroxylation difficult. The MCM-41 samples are spherical (Fig 8a and b) and the pore entrances are evenly distributed over the entire particle surface. Meanwhile, the SBA-15 samples are hexagonal and consist of a 2-D network of pores oriented along the long-axis of their geometry (Fig. 8c and d) and thus, potentially have six pore free faces where the interaction between silanol islands and surrounding trimethylsilyl groups is possible. Thus, the

potential presence of nonporous faces on the SBA-15 samples could explain the difference in the role of methylation and hydroxylation on the heterogeneous freezing abilities of the sample types at 238 K.

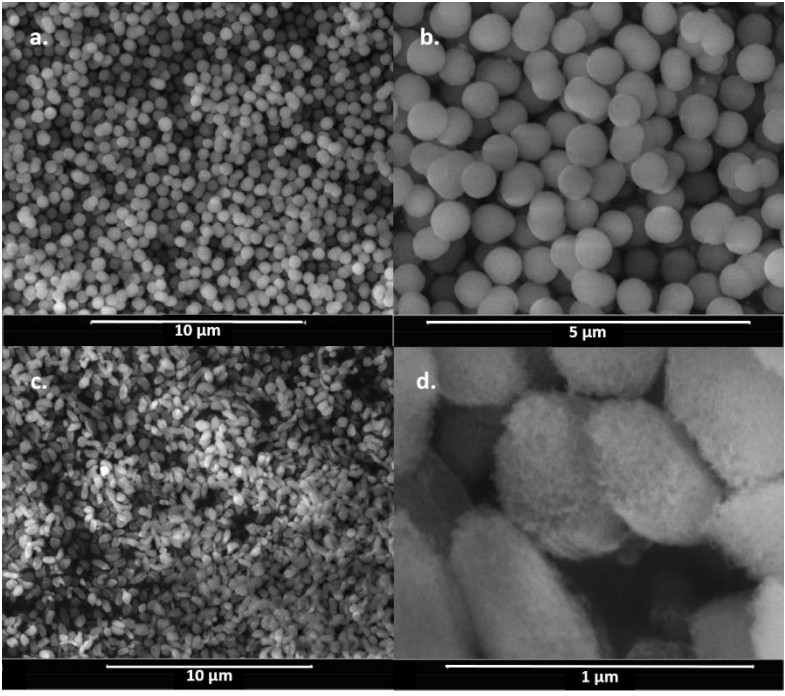

**Figure 8: Scanning election microscopy images of 2.8H2 (a. and b.) as an example of MCM-41 particles and 9.0M2 (c. and d.) as an example of SBA-15 particles.**

**4. Summary and conclusion**

In this study we have investigated the mechanism of pore condensation and freezing and its dependence on pore diameter and contact angle using synthesized silica particles with well-defined pore diameters. Particle wettability/contact angle was systematically varied through functionalizing the silica particles with hydroxyl and trimethylsilyl surface groups. The functionalized particles were characterized by $N_2$ and water vapour sorption, DRIFTS and DSC measurements. Ice nucleation

experiments on the porous particles were performed in a continuous flow diffusion chamber, covering a *T* range of 223-238 K and a relative humidity range of 100 % $RH_i$ to 105 % $RH_w$, and observed ice nucleation was compared to that predicted by PCF.



The experiments show that the presence of pores together with their diameters and contact angle are good predictors for the ice nucleation ability of particles below the HFT and below water saturation. The PCF mechanism framework accurately predicts ice nucleation at these conditions. Furthermore, the observed ice nucleation below the HFT did not support a so-called two step nucleation process confirming our previous observations using similar particle types (David et al., 2019).

Above the HFT, ice nucleation within pores cannot occur homogeneously, instead rare active sites can promote heterogeneous nucleation resulting in a lower probability of ice nucleation compared to PCF at T < HFT. Therefore, above the HFT, porosity is no longer a predictor for ice nucleation and the observed ice nucleation activity needs to be explained by the surface functionalization i.e. hydroxylation or methylation and the associated presence of so-called active sites. The enhancement in freezing due to the presence of hydroxyl or methyl groups depended on the sample type, with hydroxylated surfaces enhancing ice nucleation on the MCM-41 particles (spherical particles with pore diameters 2.4 – 3.3 nm) and methylation enhancing ice nucleation in the presence of SBA-15 particles (non-spherical of ~ 9nm pore diameter). Although the two particle types are composed of the same material, silica, the differing effect of the functional groups indicates that the role of functional groups depends on the specific surface structure. Ice nucleation at 238 K mainly occurred at water saturation, indicating that immersion or condensation freezing was the responsible mechanism.

In summary, the ability of particles to nucleate ice below water saturation at cirrus conditions can be predicted by the particle pore size distribution and contact angle. Therefore, ice nucleation parameterizations should be based on PCF below the HFT while above the HFT the presence of active sites determines the ice nucleation activity at water saturation, while the availability of active sites within pores may give rise to ice nucleation below water saturation. In the troposphere, mineral dust particles which are generally hydrophilic with low contact angles will nucleate ice via PCF and the extent of which will depend on factors such as pore size distribution and shape, contact angle and any coatings on the dust particles. Therefore, we recommend that future studies should focus on characterizing particle porosity and contact angle to better assess the role of pores on ice nucleation. We also recommend that future studies investigate the role of atmospheric aging and coatings on PCF. Certain coatings can lower the freezing point of pore water or completely block pores, inhibiting particles from nucleating ice. Thus, understanding the role of atmospheric aging on the ability of porous particles to nucleate ice via PCF is essential for understanding how anthropogenic emissions will impact future climate.

**Author Contributions**: R.O.D wrote the manuscript with contributions from C.M, Z.A.K, J.F and F.M. R.O.D conducted the ice nucleation measurements with help from F.M. R.O.D analysed the ice nucleation data. J.F and D.B synthesized and characterised the particles with varying pore diameters and functional groups. R.O.D interpreted the data with assistance from C.M and Z.A.K. C.M and Z.A.K supervised the project.





**Acknowledgements**

We would like to thank Hannes Wydler for all of his technical assistance during this project. We would also like to thank Lukas Huber at EMPA Dübendorf for performing the water sorption measurements. R.O.D, Z.A.K, D.B and J.F acknowledge support for this work from SNF grant # 200021_156581.

**Appendix A1**

The volume of water required to hydroxylate the particles ($V_w$) was calculated as:

$$V_w = \frac{m_{SiO_2} \cdot A_s \cdot \sigma_{SiOH} \cdot M_w}{\rho_W \cdot N_A} \tag{A1}$$

where $m_{SiO_2}$ is the mass of silica particles, $A_s$ is the specific surface area of the silica particles, $\sigma_{SiOH} = 4.6$ nm$^{-2}$ is the desired concentration of surface silanol groups following (Zhuralev, 2000), $M_w$ is the molar mass of water, $\rho_w$ is the density of water and $N_A$ is the Avogadro constant. The amount of organosilane added to methylate the particles was calculated following Eq. A1 except that $M_O$ and $\rho_O$ are molar mass and the density of the respective organosilane.

$$V_O = \frac{m_{SiO_2} \cdot A_s \cdot \sigma_{SiOH} \cdot M_O}{\rho_O \cdot N_A} \tag{A2}$$

The *AF* curves of the hydroxylated and methylated samples are shown in Figures A1 and A2.

**Appendix A2**

Ice crystal growth in ZINC for a given residence time ($t$), supersaturation with respect to ice ($S_i$) and temperature ($T$) was calculated based on Rogers and Yau (1989) and Lohmann et al. (2016) as follows:

$$r(t, S_i, T) = \sqrt{r_0^2 + 2\alpha \left( \frac{S_i - 1}{F_k + F_D} \right) t} \tag{A3}$$

where $r(t, S_i, T)$ is the final radius of a spherical ice crystal. A spherical assumption for ice crystals is based on observations that small ice crystals formed on spherical particles are spherical (Järvinen et al., 2016). $r_0$ represents the original radius of

the silica particles (400 nm) and is squared in the equation to account for the capacitance, which for spherical particles is equal to its radius (Rogers and Yau, 1989). $\alpha$ is the accommodation coefficient for water molecules to be incorporated into an ice lattice, which has been observed as 0.1 or 0.2 for the temperatures investigated in this study (Skrotzki et al., 2013). The terms in the denominator of Eq. A3 are:

$$F_k = \left( \frac{L_s}{R_v T} - 1 \right) \frac{L_s}{KT} \tag{A4}$$

where $L_s$ is the latent heat of sublimation as parameterized by Murphy and Koop (2005), $R_v$ is the moist gas constant and $K$ represents the thermal conductivity coefficient taken from Beard and Pruppacher, (1971).




$$F_D = \frac{R_v T}{D_v e_{s,i}(T)}$$ (A5)

Here the water vapour diffusion coefficient in air ($D_v$) was taken from Hall and Pruppacher, (1976) and $e_{s,i}(T)$ is the ice saturation vapour pressure as parameterized in Murphy and Koop, (2005).

For each experimental temperature, the $S_i$ required for a crystal to grow to 1 μm, the ice threshold in the OPC, is calculated by

5 reorganizing Eq. A3 and using a residence time of 10 s.

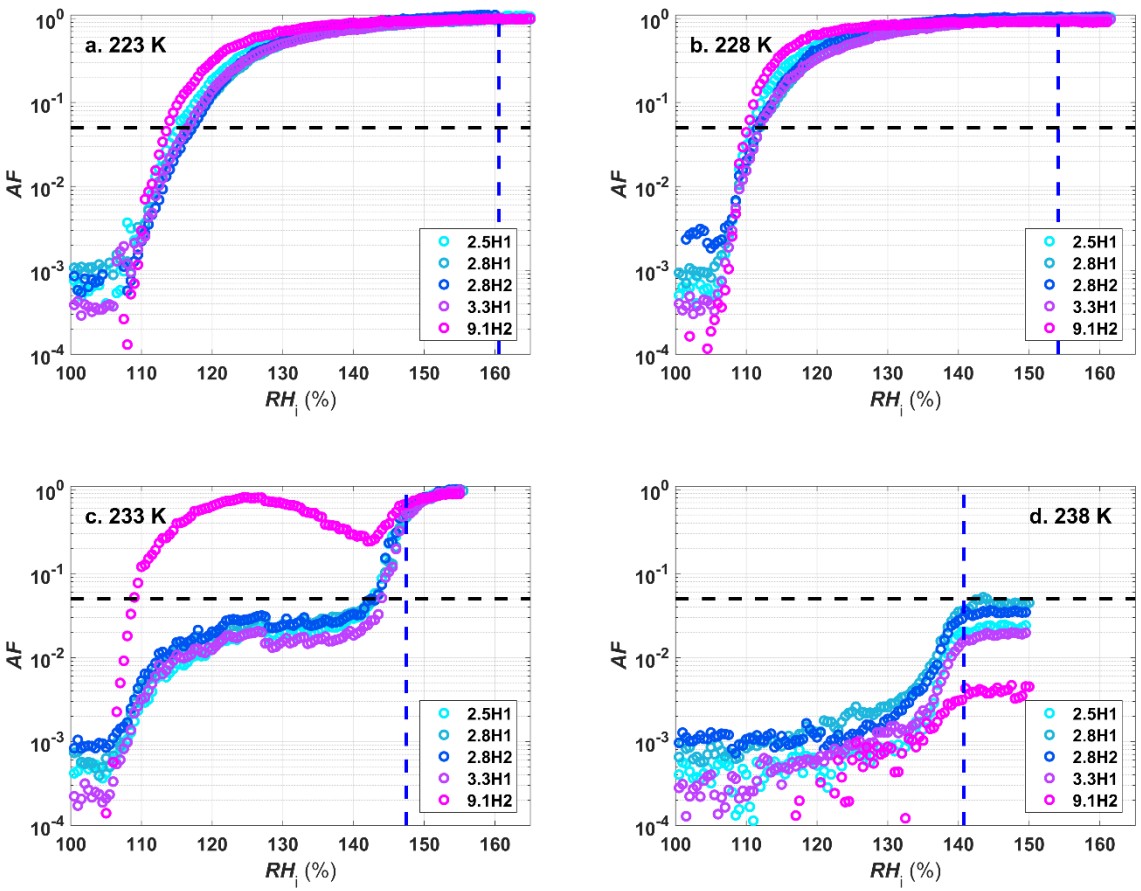

10 **Figure A1: Activated Fraction ($AF$) curves for the porous hydroxylated samples as a function of $RH_i$ at 223 K (panel a.), 228 K (panel b.), 233 K (panel c.) and 238 K (panel d.). The blue dashed vertical line represents the $RH_i$ corresponding to water saturation. The black dashed horizontal line indicates the $AF_{0.05}$ threshold. The decrease in AF for the 9.1H2 sample (magenta circles) in panel c was reproducible but disappeared at 231 K (not shown).**





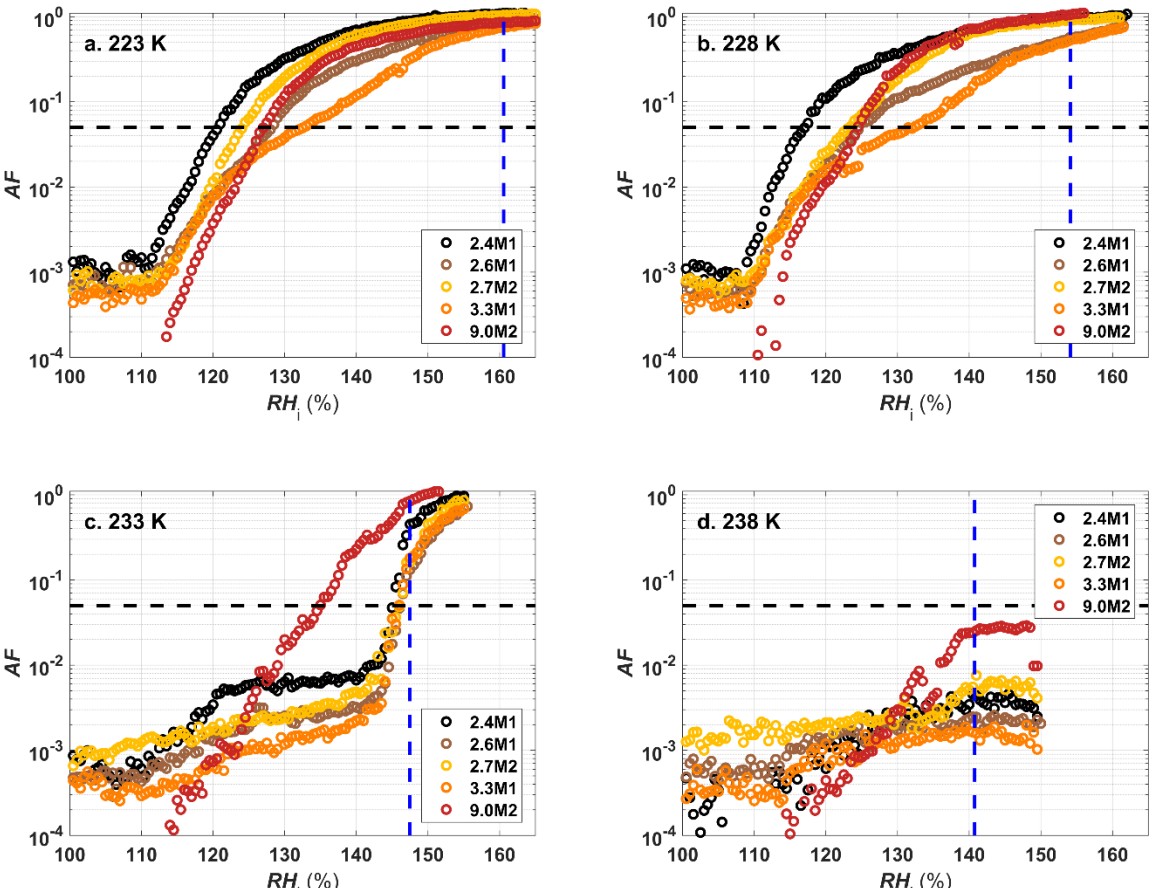

**Figure A2: Activated Fraction ($AF$) curves for the methylated samples as a function of $RH_i$ at 223 K (panel a.), 228 K (panel b.), 233 K (panel c.) and 238 K (panel d.). The blue dashed vertical line represents the $RH_i$ corresponding to water saturation. The black dashed horizontal line indicates the $AF_{0.05}$ threshold.**

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
