# Peer review of "The Role of Contact Angle and Pore Width on Pore Condensation and Freezing"

_Atmospheric Chemistry and Physics, 2019_

## Referee Comment (RC1) · Anonymous Referee #1 · 12 Dec 2019

David et al. present ice nucleation measurements with well-characterized porous silica particles to investigate the efficiency of the pore condensation and freezing (PCF) ice formation pathway as a function of the particles' pore diameter, contact angle, and surface functionalization. The reported ice nucleation behavior is generally in agreement with that predicted by the inverse Kelvin equation, which is the starting point to theoretically describe the PCF mechanism. In some cases, further considerations, like limited growth time of ice crystals in the ZINC CFDC and pressure-dependent homogeneous ice nucleation rates, were taken into account to fully explain the experimental observations. It therefore remains challenging to derive a general parameterization of the PCF mechanism for e.g. ambient dust particles, but the experiments presented in this manuscript with the synthesized particles are an important step to understand which

factors control the ice nucleation behavior of porous particles. A particular strength of the study is the comprehensive characterization of the particles with a number of different techniques as presented in Sect. 3.1. The manuscript itself is well-structured and in most parts clearly written. Reading the discussion of the ice nucleation measurements, however, was sometimes a bit challenging due to the variety of explanations taken into account to reconcile the findings with theory. I suggest below a couple of points which would need further clarifications, but otherwise recommend the publication of this article in ACP.

Specific comments:

1) Page 13, line 7: How did you exactly infer the range of contact angles for a given MCM-41 sample as also listed in Table 1 – did you use the uncertainty of d_DFT listed in Table 1, thus using e.g. pore diameters between 3 and 3.6 nm for the 3.3M1 sample in the calculation?

2) Page 13, line 12: Concerning the difficulty in the assignment of the "correct pore diameter responsible for the *initial* pore condensation": This statement seems to me a bit in contradiction with the one on page 6, line 14/15, saying that you did not use the onset relative humidity in the water sorption isotherms to compute the contact angle but the value where the pore condensation step is the steepest. So, the average pore diameters listed in Table 1 should be a reasonable choice for the computation and not the lowest value of e.g. 7 nm for the SBA-15 samples.

3) Page 14: I have a couple of questions regarding the DSC measurements: Is there an influence of external ice on the initiation of freezing in the pores (in the caption of Fig. 4 you use the expression "ice growth into pores"? Did you also record the DSC thermograms during heating, are the peak positions different from the cooling runs? How do your results compare with previous DSC measurements for MCM-41 and SBA-15 as e.g. summarized in Marcolli (2014)?

4) Page 16, line 22/23 & page 26, line 27: The reference to Skrotzki et al. (2013) does

not agree with the choice of an accommodation coefficient as low as 0.1 to compute the ice growth. As far as I can see, Skrotzki et al. (2013) reported accommodation coefficients > 0.2 for all conducted experiments with an overall average value of 0.7. So, I am not fully convinced that the limited ice growth is the best explanation for the absence of immediate ice growth as soon as ice saturation is reached in ZINC. The authors discuss on page 17 that a non-negligible fraction of the aerosol might experience a lower supersaturation than computed for the center of the aerosol lamina in the CFDC - this might also explain in part the higher onset values for ice growth on the 2.8C2 and 2.8H2 particles. Also, the onset RHi values are given for an already quite high ice-active fraction of 5%. In the full ZINC scans for 2.8H2 in Fig. A1 a and b one can see that the ice growth initiates clearly below 110% RHi – so not too far from the expected value of 100% RHi if also considering the RHi uncertainty in ZINC. Did the particles enter ZINC in a dry air flow (without pre-condensed water in the pores) – could this also influence the ice nucleation measurements?

5) Page 16, line 25 and page 26, line 24: The reference to Järvinen et al. (2016) also does not seem appropriate for me to justify a spherical shape of ice particles in the computations. They observed "quasi-spherical" ice crystals from the freezing of water droplets during strong convection, which is a clearly different process from growth of ice crystals on near-spherical solid silica particles as in the present study.

6) Page 18, line 22ff, discussion related to the increase in the AF to 0.02 for the 2.8H2 sample up to RHi = 120%. I am a bit confused by this paragraph: In the preceding paper by David et al. (2019) in PNAS, the authors show in Fig. 2 the ZINC scans for porous MCM-41 particles with 3.8 nm pore diameter. This pore diameter should be large enough to host a critical ice germ at 233 K. But in the ZINC scan at -40°C, ice formation did not occur below water saturation, and was explained by the fact that the homogeneous ice nucleation rates are not large enough to observe ice within the residence time of ZINC. If you applied the same argumentation with the pressure-dependent parameterization of CNT as presented in the current manuscript, why didn't

you detect any freezing signal for the 3.8 nm pore diameter particles at low RH? Could the small freezing signal for the 2.8H2 sample at lower RH not simply be due to immersion freezing within the pores? If I understood correctly, you used this explanation for the freezing signal observed for the 9.0M2 particles at 233 K (Fig. A2c and page 22, lines 24-27). The ZINC scan at 238 K with the 2.8H2 particles shows that at least the external particle surface contains active sites for condensation or immersion freezing (page 19, line 4/5). In contrast, the 3.8 nm pore diameter particles did not reveal a distinct immersion freezing signal at -35°C (Fig. 2 in David et al., 2019).

7) Page 21, lines 5-13: See above, would a pressure dependent parameterization of CNT not also predict a strong increase in AF for the 3.8 nm pore diameter particles at 233 K and low RH? If you also plotted the active fraction curve for the 3.8 nm pore diameter particles at 233 K from David et al. (2019) into Fig. A1c (hydroxylated and calcined samples seem to behave rather similar, see Fig. 5), the 3.8 nm sample would not show an "intermediate" ice nucleation behavior between that of the 9 nm pore diameter and that of the 2.5 – 3.3 nm pore diameter particles, but just reveal a very low ice-active fraction until reaching water saturation. How would you interpret this difference?

Minor comments and technical corrections:

1) Page 1, line 11: Maybe explicitly state "due to the inverse Kelvin effect"

2) Page 1, line 14 + 17: Avoid the repetitive statements "play an important role in determining the relative humidity" and "play an important role in predicting the humidity" in the abstract. Instead, you could also mention what type of chemical functionalization was employed and its effect on the ice nucleation behavior.

3) Page 2, line 28: "is the interfacial energy"

4) Sect. 2.1.2: You should already indicate here the larger pore sizes of the SBA-15 particles compared to MCM-41.

5) Page 5, line 4: Missing parenthesis ")" after "60°C".

6) Page 6, line 13/14: Could be phrased more clearly, you could also indicate the respective values for bulk water as a comparison.

7) Sect. 2.4: What was the relative humidity of the air before entering ZINC? Please indicate.

8) Fig. 1b & 2b: It is hard to discriminate the various bluish colors.

9) Page 12, line 4: You could also mention here in the text that you did a series of two adsorption/desorption cycles with the samples.

10) Page 12, line 19: "The relative mass . . . does not return"

11) Page 14, line 18: Maybe: "By contrast, the freezing temperature . . ."

12) Page 16, line 22: "peach" lines vs. "salmon" lines used in the caption of Fig. 5.

13) Page 17, line 3 (second line in figure caption): Could you please briefly include the factors contributing to the RHi uncertainty in ZINC?

14) Page 17, line 9: "an AF_0.05"

15) Page 21, line 25/26: Please check sentence structure.

16) Page 25, line 18 – 20: Please check sentence structure (avoid to use twice sub-clauses with "while").

---

## Referee Comment (RC2) · Anonymous Referee #2 · 4 Jan 2020

**Review of "The Role of Contact Angle and Pore Width on Pore Condensation and Freezing," by R.O. David et al. 2019**

In their submitted manuscript David et al. describe their attempts to utilize synthesized silica particles that have controlled surface pore widths, and alterations to those particles vis-á-vis functionalization, to test the utility of pore condensation freezing as a physical parameterization for ice nucleation under certain atmospheric conditions. I am impressed by the detail and depth of the described studies and the submitted manuscript. However, I do feel that there remains room for improvement and clarity in an updated manuscript before acceptance for full publication in ACP.

In particular, I suggest the authors strive to more clearly delineate where and when they think they can interpret their results purely on the basis of physical changes and where/when their understanding might be limited by the changes to the physical chemistry of the systems they are probing. In its current form the manuscript seeks to explain results primarily using physical parameters (i.e., size, geometry, bulk contact angle, etc.), but given the system (pore) sizes of exploration it might be that molecular-scale effects begin to play a role that cannot be dismissed.

Below I present an itemized list of thoughts and comments as I came to them in the text, which I hope put things into context. Following these comments I will return to some general items.

**Itemized Scientific and Editorial Comments:**

*Specific Suggestions by Page and Line Number (page, line):*

• (1,29) remove parentheses around (T)

• (1,30) I think it advisable to keep a more general definition of heterogeneous nucleation, like that of Vali et al.[1]. There he just says "nucleation aided by the presence of a foreign substance". "ice active site" could mean many things and does not necessarily give as general an impression.

• (2,7) The explanation of PCF that is when water "which can exist below water saturation in narrow pores" is confusing to me. Is it not that the local saturation condition is altered? In fact the pore water is at saturation, and this is why it condenses, it is simply not at *bulk* water saturation.

• (2, 12) I think that either "concave" should be curved, or the sign convention defined. I believe the equation is general up to the choice of sign convention for the radius of curvature.

• (2, Eq. 2) To me Eq. 2 represents an amalgamation of the critical cluster size of an ice germ in the vapor and liquid phases. Given the discussion I understand the critical radius for an ice germ in the liquid phase should be presented, and this is what is indicated by $\sigma_{iw}$ I believe. Typically $S_i$ would represent a pressure ratio $p_v/p_i$, that is the actual vapor pressure divided by the equilibrium vapor pressure over ice. In the presented case should the pressure ratio not refer to the pressure across the interface of the ice germ in liquid, i.e., $p_w/p_i$? See Marcolli[2] Eq. 11.

• (3,3) The assumption of $t = 0.38$ nm from Schrieber et al., 2001 might warrant an additional comment. It is important to note that this value will certainly be temperature dependent, and that given the changes in surface functional group it might vary between tested samples. Given that as is it represents a sizable fraction of the pore diameter such potential changes might be important. Fig.7 from Bartels-Rausch et al.[3] is a good summary of the variation over a range of temperatures that such a surface layer can take on – at the free surface, with respect to theory, measurements, and modeling. That said the pore surface is something different, beginning as a liquid/solid(pore material) interface, and when ice forms any 'quasi-liquid' layer would be a function of the specific intermolecular interactions of that system. It might be more simple to state that the expected thickness of such a layer at such temperatures would likely be of order 1 molecular layer ($\approx 2.5$Å for water).

As an additional comment related to this and Eq. (3), is that any 'quasi-liquid' layer thickness *t* would be an equilibrium phenomenon, and it is not self-evident that it should be considered as a limit to the volume available for nucleation, which is fundamentally non-equilibrium. In fact the heterogeneity at the pore surface might help to initiate nucleation even if the relaxed, low-energy state of the system would prefer disorder.

Remember taken from the point of view of pore ice the layer closest to the pore wall may be disordered or 'quasi-liquid' but taken from the viewpoint of pore water the surface layer may exhibit enhanced order. That might stimulate nucleation in analogy to what is noted (22, 25-27) in the 9.0M2 particles which the authors say contain ice nucleation sites even at higher temperatures.

- (3, 5-10) I find the discussion of water saturation in and out of the pore confusing, also see my comment on Eq. 2 above. My basic understanding is that within the pore everything is occurring in condensed liquid?

- (4,17) For the uninitiated a "Teflon bomb" is unfamiliar terminology.

- (4,25) Here and throughout the manuscript the authors should be careful with how they refer to contact angle. "these were observed to change contact angle with ageing in air" (Note aging misspelled). I believe they intend to say that the water contact angle with these was observed to change after aging in air? In any case the authors should review all uses of contact angle, or use a defined 'water contact angle' throughout, to make sure the use of contact angle is correct and consistent throughout the text.

As a follow-up question, was the changing water contact angle with calcinated particles systematic with aging as a function of any monitored variable, like RH? If so these types of particles might offer another useful experimental system.

- (5,2) "filtered off" What is meant here?

- (5,12) I think either BET should not appear as an acronym at first use, or perhaps it is enough to make it slightly more descriptive, for example: applying BET adsorption theory

- (6,14) $p/p_0$ could be better explained. It appears related to a comment below (13, 6), but the connection could be made more clear.

- (7,13) Were any other sized particles bigger or smaller than 400 nm tested? Why or why not? It would be nice to rule out any effect of particle size when considering the results.

- (7,27) Contact angle again. Here it is not the pores contact angle, but I think water's contact angle with or within the pores.

- (13,6) See also comment above with regard to how the RH of the first sorption cycle is used. As a general comment, I would expect the hysteresis between adsorption and desorption should offer an additional verification of the pore size that is presented earlier in §3.1. It seems that the magnitude of the hysteresis, if the RH ramping is done is a quasi-steady state manner, should be directly related to the stability of the liquid in the pores. Was an effort made to use the information in this manner, or do I miss a complicating factor? Finally, have the author's considered how to propagate the uncertainty to bound the uncertainty in contact angle as derived from use with Eq. 7?

- (Figure 4) The upper axes are missing a unit label. I am impressed by the agreement between the observed heat flow and the predicted critical pore diameter from the bulk physical model.

- (16,6) Why is AF 0.05 chosen? Is this simply an experimental choice of the minimum AF at which nucleation can be observed? A clear explanation would be useful, especially given that in many nucleation studies of controlled materials different AF may be chosen for plotting.

- (16, 23) Although the Skrotzki paper is directed to cirrus clouds, many such uptake measurements have been undertaken over the years and are notoriously difficult to parse. Furthermore the values in the literature vary over orders of magnitude. Direct studies of molecular uptake are presented in Kong et al.[4] as well as a review and comparison with measurement and simulation studies (including the Skrotzki paper). However, perhaps for these studies it must also be considered that the changes in functional groups that are utilized also likely lead to changes in uptake coefficient. This is clear from the adosorption/desorption isotherms in Fig. 3. It has also been previously documented that even thin surface coatings can significantly affect uptake.[5,6] I would recommend some reference to this body of work be included.

- (Fig. 7 caption) As in Fig. 6 caption it should be stated that points correspond to AF 0.05 condition.

- (Fig. 8) This figure seems a bit out of place and is of limited use in the explanation here. Perhaps it could be introduced earlier in the particle characterization section and returned to here?

- (25, 26) It might be a bit strong to say that parameterizations should be based on PCF. Perhaps, should include?

- (25, 26) The final sentence links this work to understanding anthropogenic emissions, but this is really the first mention of such emissions up to this point. Are these particles particularly analogous to any known anthropogenic emission? If the link is not strong I think this point can be left out, there are of course many reasons to better understand the effects of porosity and geometry on freezing.

**Summary:**

I have enjoyed reading the submitted manuscript. I reiterate that I think it could benefit from an improved clarity with regard to the concrete conclusions the authors would like to posit. My understanding is that basic edifice of PCF which rests on the inverse Kelvin equation does a good job of predicting the experimental observations if some of the asserted assumptions are valid. It appears that as with many systems a complete understanding of the data would involve a much more comprehensive picture of the intermolecular interactions specific to each system. The measurements rely both on ice nucleation and crystal growth to a detectible size, thus many details related to both the ice initiation and macroscopic state are convoluted. Such complexities are intrinsic in many experiments, yet I still feel this study brings the community a step forward. However, it appears the open question remains as to how this might be utilized and tested in a *messier* real atmospheric aerosol scenario.

—

[1] Vali, G., DeMott, P. J., Möhler, O., and Whale, T. F. (2015). Technical note: A proposal for ice nucleation terminology. *Atmospheric Chemistry and Physics*, 15(18):10263–10270.

[2] Marcolli, C. (2014). Deposition nucleation viewed as homogeneous or immersion freezing in pores and cavities. *Atmospheric Chemistry and Physics*, 14(4):2071–2104.

[3] Bartels-Rausch, T., Jacobi, H.-W., Kahan, T. F., Thomas, J. L., Thomson, E. S., Abbatt, J. P. D., Ammann, M., Blackford, J. R., Bluhm, H., Boxe, C., Domine, F., Frey, M. M., Gladich, I., Guzmán, M. I., Heger, D., Huthwelker, T., Klán, P., Kuhs, W. F., Kuo, M. H., Maus, S., Moussa, S. G., McNeill, V. F., Newberg, J. T., Pettersson, J. B. C., Roeselová, M., and Sodeau, J. R. (2014). A review of air–ice chemical and physical interactions (AICI): liquids, quasi-liquids, and solids in snow. *Atmospheric Chemistry and Physics*, 14(3):1587–1633.

[4] Kong, X., Papagiannakopoulos, P., Thomson, E. S., Marković, N., and Pettersson, J. B. C. (2014a). Water accommodation and desorption kinetics on ice. *The Journal of Physical Chemistry A*, 118(22):3973–3979.

[5] Kong, X., Thomson, E. S., Papagiannakopoulos, P., Johansson, S. M., and Pettersson, J. B. C. (2014b). Water accommodation on ice and organic surfaces: Insights from environmental molecular beam experiments. *The Journal of Physical Chemistry B*, 118(47):13378–13386.

[6] Thomson, E. S., Kong, X., Marković, N., Papagiannakopoulos, P., and Pettersson, J. B. C. (2013). Collision dynamics and uptake of water on alcohol-covered ice. *Atmospheric Chemistry and Physics*, 13(4):2223–2233.

---

## Author Comment (AC1) · 15 Jun 2020

**David et al. present ice nucleation measurements with well-characterized porous silica particles to investigate the efficiency of the pore condensation and freezing (PCF) ice formation pathway as a function of the particles' pore diameter, contact angle, and surface functionalization. The reported ice nucleation behavior is generally in agreement with that predicted by the inverse Kelvin equation, which is the starting point to theoretically describe the PCF mechanism. In some cases, further considerations, like limited growth time of ice crystals in the ZINC CFDC and pressure-dependent homogeneous ice nucleation rates, were taken into account to fully explain the experimental observations. It therefore remains challenging to derive a general parameterization of the PCF mechanism for e.g. ambient dust particles, but the experiments presented in this manuscript with the synthesized particles are an important step to understand which factors control the ice nucleation behavior of porous particles. A particular strength of the study is the comprehensive characterization of the particles with a number of different techniques as presented in Sect. 3.1. The manuscript itself is well-structured and in most parts clearly written. Reading the discussion of the ice nucleation measurements, however, was sometimes a bit challenging due to the variety of explanations taken into account to reconcile the findings with theory. I suggest below a couple of points which would need further clarifications, but otherwise recommend the publication of this article in ACP.**

We thank the reviewer for the positive recommendation and for raising several points that we now address individually below and in the revised manuscript to make the paper clearer.

**Specific comments:**

**1) Page 13, line 7: How did you exactly infer the range of contact angles for a given MCM-41 sample as also listed in Table 1 – did you use the uncertainty of d_DFT listed in Table 1, thus using e.g. pore diameters between 3 and 3.6 nm for the 3.3M1 sample in the calculation?**

Indeed, we used the mean $d_{DFT}$ and its uncertainty to report the spread in measured contact angles reported in Table 1 based on the humidity where the steepest condensation step was observed in the water sorption measurements. This has now been clarified on page 14 lines 7-8 by changing the sentence to read: "*The water contact angles for the MCM-41 particles ranged between 41˚- 45˚ and 75 ˚- 80˚, for the hydroxylated and methylated samples, respectively, based on the observed value and uncertainty (listed in Table 1) in the measured $d_{DFT}$.*"

**2) Page 13, line 12: Concerning the difficulty in the assignment of the "correct pore diameter responsible for the \*initial\* pore condensation": This statement seems to me a bit in contradiction with the one on page 6, line 14/15, saying that you did not use the onset relative humidity in the water sorption isotherms to compute the contact angle but the value where the pore condensation step is the steepest. So, the average pore diameters listed in Table 1 should be a reasonable choice for the computation and not the lowest value of e.g. 7 nm for the SBA-15 samples.**

Table 1 represent the bulk properties of the particles, which in this case would be representative of the majority of the pores within the particles. The narrow pore size distribution of the MCM-41 particles, allows for the steepest condensation step in the sorption measurements to be representative of the majority of the pores, especially when the uncertainty in $d_{DFT}$ is considered. Indeed, the uncertainty in $d_{DFT}$ accounted for in Table 1 and a range of possible contact angles is reported for all of the samples that underwent sorption measurements. However, as the SBA-15 particles have a wide distribution in pore diameters (7 – 16 nm, see Fig. 2), the uncertainty in $d_{DFT}$ (9 ±1.1 nm) is no longer enough to account for the range of possible pore sizes and corresponding contact angles. Furthermore, as the narrowest pores in the SBA-15 samples, which fill at lower humidities than their wider counterparts, are likely responsible for the observed freezing, it is necessary to obtain a larger range in contact angles for these particles. We have now added: "*based on the uncertainty in $d_{DFT}$ alone (±1.1 nm)*", to the sentence to clarify this point on page 15 lines 1-2.

**3) Page 14: I have a couple of questions regarding the DSC measurements: Is there an influence of external ice on the initiation of freezing in the pores (in the caption of Fig. 4 you use the expression "ice growth into pores"? Did you also record the DSC thermograms during heating, are the peak positions different from the cooling runs? How do your results compare with previous DSC measurements for MCM-41 and SBA-15 as e.g. summarized in Marcolli (2014)?**

Indeed, ice grows into the pores from the frozen bulk water in the slurry. Thus, the freezing of pore water in DSC experiments is not determined by nucleation rates but by the melting point depression within pores, as is explained on page 15 lines 6-13. Unfortunately, the heat uptake during the warming cycle was undetectable for the smaller diameter pores and is therefore not shown. For the larger diameter pores, the heat release during freezing corresponds approximately with the heat uptake during melting considering the difference in enthalpy between freezing and melting. The observed freezing temperatures are in accordance with values reported in Fig. 1 of Marcolli (2014).

**4) Page 16, line 22/23 & page 26, line 27: The reference to Skrotzki et al. (2013) does not agree with the choice of an accommodation coefficient as low as 0.1 to compute the ice growth. As far as I can see, Skrotzki et al. (2013) reported accommodation coefficients > 0.2 for all conducted experiments with an overall average value of 0.7. So, I am not fully convinced that the limited ice growth is the best explanation for the absence of immediate ice growth as soon as ice saturation is reached in ZINC.**

The reviewer is correct in that the accommodation coefficient proposed by Skrotzki et al, (2013) are above 0.2. However, Skrotzki et al, (2013) also report values from previous studies that are as low as 0.004, therefore an accommodation coefficient of 0.1 is also realistic. We have now updated the citations to cover the complete range in values reported in the literature. The following citations have been added to the main text and in the appendix: Earle et al., (2010), Isono and Iwai, (1969) and Magee et al., (2006). Therefore, we still believe that a low accommodation coefficient is a valid explanation for the delayed onset and keep the lines for 0.2 and 0.1 as accommodation coefficients in Figure 6 and Figure 6, 7, and 8, respectively.

**The authors discuss on page 17 that a non-negligible fraction of the aerosol might experience a lower supersaturation than computed for the center of the aerosol lamina in the CFDC - this might also explain in part the higher onset values for ice growth on the 2.8C2 and 2.8H2 particles.**

Yet, we agree with the reviewer that a low accommodation coefficient is not the only possible explanation for the absence of immediate ice growth. The increase in humidity above 100% for first ice crystal detection could also be due to the initial ice growth out of the pores being stacking disordered. As has been shown in molecular dynamic simulations, initial ice nucleation and ice in confinement is typically stacking disordered (e.g. Lupi et al., 2017; Moore et al., 2010). Stacking disordered ice requires a higher humidity to be stable than bulk ice. As such, a higher humidity would be required for the ice to initially grow. However, once the first few monolayers of ice covers the particle surface, ice growth likely transitions to hexagonal ice. If this transition takes longer than some seconds, ice growth within ZINC would require supersaturation with respect to bulk ice, since the growth is occurring on stacking disordered ice. Additionally, it is possible that some of the particles are exposed to lower humidities than expected, if they are traveling outside of the lamina (Garimella et al., 2017). However, this effect is more relevant at higher humidities when the gradient in temperature between the cold and warm wall is higher. Therefore, we do not expect this to be a major issue at these low humidities.

**Also, the onset RHi values are given for an already quite high ice-active fraction of 5%. In the full ZINC scans for 2.8H2 in Fig. A1 a and b one can see that the ice growth initiates clearly below 110% RHi – so not too far from the expected value of 100% RHi if also considering the RHi uncertainty in ZINC.**

Indeed, as mentioned by the reviewer, at these humdities, the uncertainty due to the temperature reported by the thermocouples and thus the calculated relative humidity that the particles are exposed to (~ ± 5% $RH_i$) presents an additional uncertainty in our reported onsets. As such it is possible that the observed onsets (Fig. A1) could be occurring closer to ice saturation than we show considering the 5% uncertainty in $RH_i$. We revised the manuscript to discuss all these potential explanations in the text by stating the following:

*"Moreover, the reported AF of 0.05 do not correspond with the ice onset. Considering Fig. A1, the initial ice is observed at $RH_i$ 110 % and 108% for 223 and 228 K, respectively. Lupi et al., (2017) and Moore et al., (2010) have shown in MD studies that stacking disordered ice is formed in confinement and during nucleation, requiring a humidity higher than 100% to grow ice. Additionally, the calculated humidity that particles are exposed to in ZINC depends on the temperatures of the warm and cold walls, which are measured by thermocouples that have an uncertainty of ±0.1 K (Stetzer et al., 2008). This uncertainty (±5 %) can lead to a higher reported humidity than required to observe ice nucleation and is included in the vertical error bars in Figure 6, 7 and 8."* on page 18 lines 22-29.

**Did the particles enter ZINC in a dry air flow (without pre-condensed water in the pores) – could this also influence the ice nucleation measurements?**

Additionally, we have now mentioned in the text that the particles were aerosolized and transported to ZINC using evaporated liquid nitrogen on page 8 lines 5-6.

**5) Page 16, line 25 and page 26, line 24: The reference to Järvinen et al. (2016) also does not seem appropriate for me to justify a spherical shape of ice particles in the computations. They observed "quasi-spherical" ice crystals from the freezing of water droplets during strong convection, which is a clearly different process from growth of ice crystals on near-spherical solid silica particles as in the present study.**

We agree with the reviewer's opinion that the freezing and subsequent growth of supercooled droplets is likely a different process than the growth of ice on spherical aerosol particles. We have therefore changed the citation to Harrington et al., (2019), where it is assumed that at such small ice crystal sizes (~1 μm), the ice is spherical. We have now

updated the sentence to read: *"The ice crystal shape in the growth calculation was assumed to be spherical due to the small final size (~1 µm) and its growth on spherical particles (Harrington et al., 2019)."* on page 18 lines 19-20.

**6) Page 18, line 22ff, discussion related to the increase in the AF to 0.02 for the 2.8H2 sample up to RHi = 120%. I am a bit confused by this paragraph: In the preceding paper by David et al. (2019) in PNAS, the authors show in Fig. 2 the ZINC scans for porous MCM-41 particles with 3.8 nm pore diameter. This pore diameter should be large enough to host a critical ice germ at 233 K. But in the ZINC scan at -40°C, ice formation did not occur below water saturation, and was explained by the fact that the homogeneous ice nucleation rates are not large enough to observe ice within the residence time of ZINC. If you applied the same argumentation with the pressure dependent parameterization of CNT as presented in the current manuscript, why didn't you detect any freezing signal for the 3.8 nm pore diameter particles at low RH? Could the small freezing signal for the 2.8H2 sample at lower RH not simply be due to immersion freezing within the pores? If I understood correctly, you used this explanation for the freezing signal observed for the 9.0M2 particles at 233 K (Fig. A2c and page 22, lines 24-27). The ZINC scan at 238 K with the 2.8H2 particles shows that at least the external particle surface contains active sites for condensation or immersion freezing (page 19, line 4/5). In contrast, the 3.8 nm pore diameter particles did not reveal a distinct immersion freezing signal at -35°C (Fig. 2 in David et al., 2019).**

The width of the pore size distribution and the dependence of nucleation rates and critical embryo size on CNT parameterization hamper the precise analysis of how freezing of pore water and growth out of pores depends on pore width and pore water volume. Because of these uncertainties, it is difficult to come to exact conclusions whether nucleation rates or pore diameter limit freezing within pores at 233 K. Applying a pressure dependent nucleation rate, ice in the 3.8 nm pores is expected to freeze at 233 K at negative pressures and within the residence time of ZINC, as the pores should be wide enough to host ice. However, since we did not observe freezing, the increase in the nucleation rate due to negative pressure is likely not the reason or counteracted by some other effect. We explain this in the following.

First, we consider immersion freezing as an alternative explanation for the weak freezing signal below water saturation for the sample 2.8H2 (see Fig. A1c). Yet even if there were nucleation sites within the pores, the DSC measurements indicate that the pores should be too narrow to host ice at 233 K (see Fig 5b) and the immersion sites are therefore, ineffective for ice nucleation. However, we cannot rule out that there are some pores large enough to accommodate the critical ice germ in the 2.5-2.8 nm samples where the freezing is not detectable in the DSC measurements. Indeed, the pore size distributions shown in Figure 2 indicate that these samples do have some pores larger than 3 nm in diameter, which is close to the critical ice germ size at 233 K depending on the parametrization used (for example with the Murray et al, (2010) parametrization for hexagonal ice, the critical ice germ should be 2.9 nm at 233 K assuming a quasi-liquid layer of 0.38 nm). As such, it is possible that some pores in a small fraction of the particles are capable of supporting ice that freezes either homogeneously due to negative pressure or heterogeneously due to ice active sites within some of the pores. As to why we do not observe this freezing in the 3.8 nm samples used in David et al. (2019) we cannot fully isolate one reason. It may be because the particles in David et al. (2019) were not hydroxylated like the ones in this work (i.e. they were calcined), the water contact angle with the pore wall is expected to be higher. If this is the case, the pore fills at a higher relative humidity when the curvature of the pore water is less and the water would therefore experience a lower negative pressure in these wider pores, thus decreasing the homogeneous nucleation rate due to negative pressure in these pores.

As the reviewer states, since the 3.8 nm samples did not exhibit any heterogeneous freezing ability (at T = 238 K) in David et al. (2019), then it is likely that no active sites exist in the pores even at 233 K. The combination of no active sites and a low nucleation rate from the lower negative pressure may be the reason no freezing was observed in the 3.8 nm samples in David et al. (2019).

**7) Page 21, lines 5-13: See above, would a pressure dependent parameterization of CNT not also predict a strong increase in AF for the 3.8 nm pore diameter particles at 233 K and low RH? If you also plotted the active fraction curve for the 3.8 nm pore diameter particles at 233 K from David et al. (2019) into Fig. A1c (hydroxylated and calcined samples seem to behave rather similar, see Fig. 5), the 3.8 nm sample would not show an "intermediate" ice nucleation behavior between that of the 9 nm pore diameter and that of the 2.5 – 3.3 nm pore diameter particles, but just reveal a very low ice-active fraction until reaching water saturation. How would you interpret this difference?**

Here again, the width of the pore size distribution and the dependence of nucleation rates and critical embryo size on CNT parameterization hamper a precise analysis. Comparing the pore size distribution of the 3.8 nm sample investigated in David et al. (2019, Fig.S5) and of 3.3H1 (Fig. 1b), there is a large overlap in pore diameters and especially the tail to large pores reaches for both samples up to 4.5 nm. If the pore width is limiting the stability of ice within pores, these pores indeed should behave similarly. However, as mentioned in the previous response, the negative pressure in the 3.8 nm pores in the samples from David et al, (2019) may differ even from the calcined samples (2.8C2) examined in this study due to the unstable nature of calcined samples depending on their age and exposure to water vapour (Muster et al., 2001). Therefore, it is not straightforward to extrapolate the results from the current study to those of David et al. (2019).

**Minor comments and technical corrections:**

**1) Page 1, line 11: Maybe explicitly state "due to the inverse Kelvin effect"**

We have now rephrased the sentence to state "*due to the inverse Kelvin effect.*" (see page 1 line 13)

**2) Page 1, line 14 + 17: Avoid the repetitive statements "play an important role in determining the relative humidity" and "play an important role in predicting the humidity" in the abstract. Instead, you could also mention what type of chemical functionalization was employed and its effect on the ice nucleation behavior.**

We remove the sentence referenced by the reviewer and shorten the sentence to read (see page 1 lines 16-18):

"*We find that for the pore diameters (2.2 – 9.2 nm) and water contact angles (15 – 78˚) covered in this study, our results reveal that the water contact angle plays an important role in predicting the humidity required for pore filling while the pore diameter determines the ability of pore water to freeze.*"

**3) Page 2, line 28: "is the interfacial energy"**

Added, thanks.

**4) Sect. 2.1.2: You should already indicate here the larger pore sizes of the SBA-15 particles compared to MCM-41.**

We have now mentioned that larger pores were achieved using SBA-15 stating: "*To obtain larger pore diameters (~9 nm), SBA-15 particles (see Fig. 1c and d.) were synthesized similarly to Linton et al., (2009b) where…*" on page 4 line 18.

**5) Page 5, line 4: Missing parenthesis ")" after "60◦C".**

Added, thanks

**6) Page 6, line 13/14: Could be phrased more clearly, you could also indicate the respective values for bulk water as a comparison.**

We have rephrased the sentence to read "*For water in confinement at 25 ˚C, the values of $\gamma(T)$ and $v_l(T)$ are 71.69 mN/m and 20.5 $m^3$/mol, respectively (Kocherbitov and Alfredsson, 2007).*" on page 7 lines 9-10. We do not discuss the values for bulk water as it is beyond the scope of the current study.

**7) Sect. 2.4: What was the relative humidity of the air before entering ZINC? Please indicate.**

The humidity of the air entering ZINC is close to zero (< 1 %) as evaporated liquid nitrogen was used to aerosolize and transport the particles to ZINC. We have now clarified this in the text by stating: "*…supplied with evaporated liquid nitrogen (purity 6.0) to eliminate any residual humidity (RH < 1 % at 223 K)…*" on page 8 lines 5-6.

**8) Fig. 1b & 2b: It is hard to discriminate the various bluish colors.**

We thank the reviewer for pointing this out. We have now increased the resolution of the figure, which makes the color scheme more distinguishable (now Fig. 2b and 3b).

**9) Page 12, line 4: You could also mention here in the text that you did a series of two adsorption/desorption cycles with the samples.**

We have updated the text to read: "*Two water vapour sorption cycles were obtained for the samples 2.4M1, 2.5H1, 3.3M1, 3.3H1, 9.1H2 and 9.0M2 and the resultant isotherms are shown in Fig. 4*" on page 13 line 4-5.

**10) Page 12, line 19: "The relative mass . . . does not return"**

Corrected, thanks

**11) Page 14, line 18: Maybe: "By contrast, the freezing temperature . . ."**

We have updated the text to read: "*In contrast,…*" on page 15 line 20

**12) Page 16, line 22: "peach" lines vs. "salmon" lines used in the caption of Fig. 5.**

Thank you, we have updated to read "salmon" throughout the text.

**13) Page 17, line 3 (second line in figure caption): Could you please briefly include the factors contributing to the RHi uncertainty in ZINC?**

Thank you, we have now reworded the sentence to read: "*The error bars represent the maximum uncertainty in the calculated $RH_i$ (±5 %) in ZINC arising from the uncertainty in the reported thermocouple temperature (±0.1 K; Stetzer et al., 2008) and encompass the standard deviation from averaging the experiments.*" in the figure caption

**14) Page 17, line 9: "an AF_0.05"**

Updated, thanks.

**15) Page 21, line 25/26: Please check sentence structure.**

We have updated the sentence to read: "*At 223 and 228 K, the samples with 2.4 nm pore diameter had the lowest $AF_{0.05}$ $RH_i$ and the 3.3 nm particles the highest. The 2.6, 2.7, and 9.0 nm samples are in-between and overlap (Fig. 8).*" on page 23 lines 31-33.

**16) Page 25, line 18 – 20: Please check sentence structure (avoid to use twice subclauses with "while").**

Thank you for pointing this out. We have now split the sentence to read as follows: "*Therefore, ice nucleation parameterizations should be based on PCF below the HFT. Above the HFT, active sites present on the particle surface determine the ice nucleation activity at water saturation, while below water saturation active sites within pores are required to nucleate ice.*" on page 27 lines 5-7.

**Author References**

Earle, M. E., Kuhn, T., Khalizov, A. F. and Sloan, J. J.: Volume nucleation rates for homogeneous freezing in supercooled water microdroplets: results from a combined experimental and modelling approach, Atmospheric Chem. Phys., 10(16), 7945–7961, doi:https://doi.org/10.5194/acp-10-7945-2010, 2010.

Garimella, S., Rothenberg, D. A., Wolf, M. J., David, R. O., Kanji, Z. A., Wang, C., Rösch, M. and Cziczo, D. J.: Uncertainty in counting ice nucleating particles with continuous flow diffusion chambers, Atmos Chem Phys, 17(17), 10855–10864, doi:10.5194/acp-17-10855-2017, 2017.

Harrington, J. Y., Moyle, A., Hanson, L. E. and Morrison, H.: On Calculating Deposition Coefficients and Aspect-Ratio Evolution in Approximate Models of Ice Crystal Vapor Growth, J. Atmospheric Sci., 76(6), 1609–1625, doi:10.1175/JAS-D-18-0319.1, 2019.

Isono, K. and Iwai, K.: Growth Mode of Ice Crystals in Air at Low Pressure, Nature, 223(5211), 1149–1150, doi:10.1038/2231149a0, 1969.

Kocherbitov, V. and Alfredsson, V.: Hydration of MCM-41 Studied by Sorption Calorimetry, J. Phys. Chem. C, 111(35), 12906–12913, doi:10.1021/jp072474r, 2007.

Linton, P., Hernandez-Garrido, J.-C., Midgley, P. A., Wennerström, H. and Alfredsson, V.: Morphology of SBA-15-directed by association processes and surface energies, Phys. Chem. Chem. Phys., 11(46), 10973–10982, doi:10.1039/B913755F, 2009.

Lupi, L., Hudait, A., Peters, B., Grünwald, M., Mullen, R. G., Nguyen, A. H. and Molinero, V.: Role of stacking disorder in ice nucleation, Nature, 551(7679), nature24279, doi:10.1038/nature24279, 2017.

Magee, N., Moyle, A. M. and Lamb, D.: Experimental determination of the deposition coefficient of small cirrus-like ice crystals near −50°C, Geophys. Res. Lett., 33(17), doi:10.1029/2006GL026665, 2006.

Moore, E. B., de la Llave, E., Welke, K., Scherlis, D. A. and Molinero, V.: Freezing, melting and structure of ice in a hydrophilic nanopore, Phys. Chem. Chem. Phys., 12(16), 4124, doi:10.1039/b919724a, 2010.

Murray, B. J., L. Broadley, S., W. Wilson, T., J. Bull, S., H. Wills, R., K. Christenson, H. and J. Murray, E.: Kinetics of the homogeneous freezing of water, Phys. Chem. Chem. Phys., 12(35), 10380–10387, doi:10.1039/C003297B, 2010.

Muster, T. H., Prestidge, C. A. and Hayes, R. A.: Water adsorption kinetics and contact angles of silica particles, Colloids Surf. Physicochem. Eng. Asp., 176(2), 253–266, 2001.

Stetzer, O., Baschek, B., Lüönd, F. and Lohmann, U.: The Zurich Ice Nucleation Chamber (ZINC)-A New Instrument to Investigate Atmospheric Ice Formation, Aerosol Sci. Technol., 42(1), 64–74, doi:10.1080/02786820701787944, 2008.

---

## Author Comment (AC2) · 15 Jun 2020

Reviewer comments are reproduced in bold and author responses are in regular typeface. All line numbers in the authors' response refer to the revised manuscript. Revised text is given in italics.

**Review of "The Role of Contact Angle and Pore Width on Pore Condensation and Freezing," by R.O. David et al. 2019**

**Anonymous Referee #2**

**In their submitted manuscript, David et al. describe their attempts to utilize synthesized silica particles that have controlled surface pore widths, and alterations to those particles vis-a-vis functionalization, ´ to test the utility of pore condensation freezing as a physical parameterization for ice nucleation under certain atmospheric conditions. I am impressed by the detail and depth of the described studies and the submitted manuscript. However, I do feel that there remains room for improvement and clarity in an updated manuscript before acceptance for full publication in ACP. In particular, I suggest the authors strive to more clearly delineate where and when they think they can interpret their results purely on the basis of physical changes and where/when their understanding might be limited by the changes to the physical chemistry of the systems they are probing. In its current form the manuscript seeks to explain results primarily using physical parameters (i.e., size, geometry, bulk contact angle, etc.), but given the system (pore) sizes of exploration it might be that molecular-scale effects begin to play a role that cannot be dismissed.**

**Below I present an itemized list of thoughts and comments as I came to them in the text, which I hope put things into context. Following these comments I will return to some general items.**

We thank the reviewer for the positive recommendation and for raising several points that we now address individually below and in the revised manuscript to make the paper clearer.

**Itemized Scientific and Editorial Comments:**

*Specific Suggestions by Page and Line Number (page, line):*

**• (1,29) remove parentheses around (T)**

Removed, thank you

**• (1,30) I think it advisable to keep a more general definition of heterogeneous nucleation, like that of Vali et al. 1 . There he just says "nucleation aided by the presence of a foreign substance". "ice active site" could mean many things and does not necessarily give as general an impression.**

We have reformulated to be consistent with the description provided by Vali et al. (2015*): At T > HFT ice formation takes place heterogeneously and is aided by the presence of a foreign substance (Fletcher, 1969; Kaufmann et al., 2017; Kiselev et al., 2017; Vali et al., 2015), which lowers the energy barrier required for the homogeneous nucleation of ice.*

**• (2,7) The explanation of PCF that is when water "which can exist below water saturation in narrow pores" is confusing to me. Is it not that the local saturation condition is altered? In fact the pore water is at saturation, and this is why it condenses, it is simply not at bulk water saturation.**

That is correct, as such we have revised the sentence to read: "*PCF occurs when liquid water, which can exist in narrow pores, cracks, cavities or capillaries (hereafter referred to as pores) below ambient water saturation, freezes.*" on page 2 lines 8-9.

**• (2, 12) I think that either "concave" should be curved, or the sign convention defined. I believe the equation is general up to the choice of sign convention for the radius of curvature.**

For clarity we have now reformulated the sentence to state "*negative or concave curvature*", on page 2, line 9.

**• (2, Eq. 2) To me Eq. 2 represents an amalgamation of the critical cluster size of an ice germ in the vapor and liquid phases. Given the discussion I understand the critical radius for an ice germ in the liquid phase should be presented, and this is what is indicated by σiw I believe. Typically Si would represent a pressure ratio pv/pi , that is the actual vapor pressure divided by the equilibrium vapor pressure over ice. In the presented case should the pressure ratio not refer to the pressure across the interface of the ice germ in liquid, i.e., pw/pi? See Marcolli 2 Eq. 11.**

Thank you for pointing this out, indeed the ratio should be that of water and ice. The equation has been updated accordingly. The description of Eq. 2 now reads: "*where $\sigma_{iw}$ is the interfacial energy between the ice and water interface, $v_{ice}$ is the approximate volume of bulk ice, and $\frac{p_w}{p_i}$ is the ratio of the equilibrium vapour pressures over water and ice.*" on page 2 lines 29-30.

**• (3,3) The assumption of t = 0.38 nm from Schrieber et al., 2001 might warrant an additional comment. It is important to note that this value will certainly be temperature dependent, and that given the changes in surface functional group it might vary between tested samples. Given that as is it represents a sizable fraction of the pore diameter such potential changes might be important. Fig.7 from Bartels-Rausch et al. 3 is a good summary of the variation over a range of temperatures that such a surface layer can take on – at the free surface, with respect to theory, measurements, and modeling. That said the pore surface is something different, beginning as a liquid/solid(pore material) interface, and when ice forms any 'quasi-liquid' layer would be a function of the specific intermolecular interactions of that system. It might be more simple to state that the expected thickness of such a layer at such temperatures would likely be of order 1 molecular layer (≈ 2.5A for water).**

Indeed, the width of the quasi-liquid layer on flat surfaces varies with temperature and with surface functional groups, and the same is the case for the quasi-liquid layer within pores. Yet, the exact width of the quasi-liquid layer is uncertain and a parameterization as a function of temperature out of scope. Schreiber et al. (2001) determined a value of $t = 0.38$ nm by fitting measured melting point depressions to a modified Gibbs-Thomson equation. Since the temperature range (223 - 263 K) and pore diameters (3 – 12 nm) investigated in Schreiber et al, (2001) are the same as in this study, $t = 0.38$ nm seems appropriate for the experiments carried out in this study. Furthermore, the assumption of $t = 0.38$ nm leads to a good agreement with our own results as can be seen from the DSC measurements and with melting point depressions compiled from literature in Marcolli (2014). We have now added: "*The width of the quasi-liquid layer has been shown to depend on temperature and surface chemistry but the exact thickness of the layer varies greatly between different observational techniques and MD studies (Bartels-Rausch et al., 2014). Nevertheless, the thickness of the quasi-liquid layer can be parameterized by fitting the measured melting point depressions of ice in pores to a modified version of the Gibbs-Thomson equation and has been shown to*

*vary between 1 and 2 monolayers thick for the pore diameters and across the temperature range investigated in this study* (Findenegg et al., 2008; Jähnert et al., 2008; Marcolli, 2014; Schreiber et al., 2001; Wang et al., 2019). *When accounting for the quasi-liquid layer thickness, assumed as t = 0.38 nm (Schreiber et al., 2001), the diameter of a pore capable of hosting ice ($D_p$) can be expressed as:…*" to the discussion on page 3 lines 1-9.

**As an additional comment related to this and Eq. (3), is that any 'quasi-liquid' layer thickness t would be an equilibrium phenomenon, and it is not self-evident that it should be considered as a limit to the volume available for nucleation, which is fundamentally non-equilibrium. In fact the heterogeneity at the pore surface might help to initiate nucleation even if the relaxed, low-energy state of the system would prefer disorder.**

**Remember taken from the point of view of pore ice the layer closest to the pore wall may be disordered or 'quasi-liquid' but taken from the viewpoint of pore water the surface layer may exhibit enhanced order. That might stimulate nucleation in analogy to what is noted (22, 25-27) in the 9.0M2 particles which the authors say contain ice nucleation sites even at higher temperatures.**

The water in the quasi-liquid layer is not available for incorporation into the critical ice embryo as it is bound to the pore surface (i.e. due to the interaction with the pore surface, it is at a lower chemical potential than the bulk of the pore water). Since the interaction with the pore wall depends on the functionalization of the pore surface, the thickness and the structure of the quasi-liquid layer indeed depends on the pore surface properties and it also may be structured in a way suitable for ice formation, i.e. it may act as an ice-nucleating surface inducing heterogeneous ice nucleation. Within the framework of CNT, this is described by amending the Gibbs free energy barrier of nucleation with a term that depends on the contact angle between the ice embryo and the pore surface, implying a spherical-cap-shaped embryo instead of a sphere. However, heterogeneous ice nucleation would still be limited by the width of the pore and therefore, the pore diameter would still need to be wide enough for bulk ice to fit inside the pore. For the pore diameters where heterogeneous freezing can occur without being impeded by the critical pore diameters (9.1H2 and 9.0M2) the ordering/disordering of the quasi-liquid layer may be responsible for the observed freezing. We discuss this on page 25 starting on line 10.

**• (3, 5-10) I find the discussion of water saturation in and out of the pore confusing, also see my comment on Eq. 2 above. My basic understanding is that within the pore everything is occurring in condensed liquid?**

We have reformulated the paragraph for clarity and it now reads: "*Based on CNT, the ice growing out of the pore needs to be supercritical with respect to the vapour phase. The energy barrier for nucleation in the vapour phase is significantly higher than that in water. This increase in energy barrier comes from the need to replace $\sigma_{iw}$ with the interfacial energy between ice and vapour ($\sigma_{iv}$) in Eq. 2, which is approximately a factor of 4.8 larger than $\sigma_{iw}$ at 236 K (Cooper, 1974; Ickes et al., 2015; Ketcham and Hobbs, 1969). Additionally, as the ice growing out of the pore experiences an environment that is subsaturated with respect to water, $\frac{p_w}{p_i}$ in Eq. 2 must be replaced by the ice saturation ratio ($S_i$), which is smaller than $\frac{p_w}{p_i}$. Therefore, the critical radius for ice growth out of the pore is much larger than that of the critical radius in the pore necessitating a substantial increase in the ice saturation ratio ($S_i$) for ice to be able to grow out of a pore…*" on page 3 lines 12-19.

**• (4,17) For the uninitiated a "Teflon bomb" is unfamiliar terminology.**

Thank you for pointing this out. We have now revised the terminology to state: "*a Teflon lined acid digestion vessel*" on page 4, lines 14-15

**• (4,25) Here and throughout the manuscript the authors should be careful with how they refer to contact angle. "these were observed to change contact angle with ageing in air" (Note aging misspelled).**

Thank you for pointing this out. We have now reformulated the sentence to state: "*We will focus on ice nucleation experiments with particles functionalized with trimethyl and hydroxyl groups rather than just calcined ones, as their contact angle with water was observed to change with aging in air (Muster et al., 2001).*" on page 5 lines 6-8.

**I believe they intend to say that the water contact angle with these was observed to change after aging in air? In any case the authors should review all uses of contact angle, or use a defined 'water contact angle' throughout, to make sure the use of contact angle is correct and consistent throughout the text.**

We have now revised the manuscript to clarify that contact angle refers to the water contact angle.

**As a follow-up question, was the changing water contact angle with calcinated particles systematic with aging as a function of any monitored variable, like RH? If so these types of particles might offer another useful experimental system.**

Unfortunately, the storage humidity was not monitored over the period during which we performed experiments with calcined particles. Therefore, it is not possible to quantify the time or conditions that led to observed changes in the water contact angle of the particles. As such, those experiments are not included in the present manuscript. Nevertheless, this may be a valid approach to investigate the impact of water contact angles on the PCF mechanism for future studies.

**• (5,2) "filtered off" What is meant here?**

We have clarified the text to read: "… *before the suspension was filtered and washed*…" on page 5 line 16

**• (5,12) I think either BET should not appear as an acronym at first use, or perhaps it is enough to make it slightly more descriptive, for example: applying BET adsorption theory**

Thanks, we have now changed the text to read: "… *and applying the Brunauer, Emmett and Teller (BET) gas adsorption theory*…" on page 6 lines 7-8

**• (6,14) $p/p_0$ could be better explained. It appears related to a comment below (13, 6), but the connection could be made more clear.**

To connect $p/p_0$ to $RH_w$ we have changed the text to read: "…*$p/p_0$ is the water saturation ratio or $RH_w/100$.*" On page 7 line 2. Additionally, we have clarified the $p/p_0$ value used when calculating the water contact angle of the pores by stating: "*When deriving $\theta$, $p/p_0$ is identified as the saturation ratio where the pore condensation step of the DVS measurement is the steepest.*" on page 7 line 10-11.

**• (7,13) Were any other sized particles bigger or smaller than 400 nm tested? Why or why not? It would be nice to rule out any effect of particle size when considering the results.**

The particles were synthesized to be 400 nm in order for the observed differences in ice nucleation ability to be directly attributable to differences in pore diameters and water contact angles of the particles rather than the particle size. I. e. it was a means to eliminate one variable. The effect of particle size might be an additional parameter to be tested in future studies.

**• (7,27) Contact angle again. Here it is not the pores contact angle, but I think water's contact angle with or within the pores.**

In accordance with the previous comment on contact angle, we have now updated the manuscript to clearly indicate that the water contact angle with the pore or particle surface is meant.

**• (13,6) See also comment above with regard to how the RH of the first sorption cycle is used. As a general comment, I would expect the hysteresis between adsorption and desorption should offer an additional verification of the pore size that is presented earlier in §3.1. It seems that the magnitude of the hysteresis, if the RH ramping is done is a quasi-steady state manner, should be directly related to the stability of the liquid in the pores. Was an effort made to use the information in this manner, or do I miss a complicating factor? Finally, have the author's considered how to propagate the uncertainty to bound the uncertainty in contact angle as derived from use with Eq. 7?**

The observed hysteresis between adsorption and desorption isotherms can indeed be used to infer the pore shape and pore size as long as the pores maintain a constant water contact angle. However, the change in adsorption and desorption isotherms in the subsequent cycles indicates that the water contact angle of the pores changes with initial exposure to increasing humidity. We therefore use nitrogen adsorption to determine the pore diameters and the initial water vapour sorption cycle to infer contact angles using the pore diameters determined from nitrogen adsorption. The water uptake during the first water vapour sorption cycle is most representative of the contact angle relevant for water uptake in our ice nucleation experiments as the experiments are the first instance that the pores are exposed to high concentrations of water vapor.

**• (Figure 4) The upper axes are missing a unit label. I am impressed by the agreement between the observed heat flow and the predicted critical pore diameter from the bulk physical model.**

Thank you, we have now added "*nm*" to the upper axis of this figure. Indeed, it shows that the values used to constrain the interfacial tension from Murray et al., (2010) and Zobrist et al., (2007) are working well. Additionally, it corroborates that assuming a quasi-liquid layer of 0.38 nm is appropriate.

**• (16,6) Why is AF 0.05 chosen? Is this simply an experimental choice of the minimum AF at which nucleation can be observed? A clear explanation would be useful, especially given that in many nucleation studies of controlled materials different AF may be chosen for plotting.**

Thank you for pointing this out. The *AF* of 0.05 was chosen due to the distribution in pore sizes and functional groups on the particles. Using a lower AF to represent the average freezing RH of the porous particles, may lead to the misinterpretation of the freezing results due to the presence of unique particle features that may exist on a few of the particles. We have now added the following to clarify this choice in the main text: "*An AF of 0.05 was chosen as best representing the average freezing RH of the porous particles.*" starting on page 17 line 10.

**• (16, 23) Although the Skrotzki paper is directed to cirrus clouds, many such uptake measurements have been undertaken over the years and are notoriously difficult to parse. Furthermore the values in the literature vary over orders of magnitude. Direct studies of molecular uptake are presented in Kong et al. 4 as well as a review and comparison with measurement and simulation studies (including the Skrotzki paper). However, perhaps for these studies it must also be considered that the changes in functional groups that are utilized also likely lead to changes in uptake coefficient. This is clear from the adosorption/desorption isotherms in Fig. 3. It has also been previously documented that even thin surface coatings can significantly affect uptake.5,6 I would recommend some reference to this body of work be included.**

Since pores are closely spaced in our particles, pore openings make up a relevant fraction of the particle surface. We therefore assume that ice grows on the ice-covered part of the particle surface, for which the accommodation coefficient on ice is relevant. Molecular dynamic simulations presented in David et al. (2019), which show that ice rapidly covers the entire particle surface, further support this assumption.

We revised the discussion of the delayed freezing onset RH following a comment of reviewer #1, who pointed out that an accommodation coefficient of 0.1 is at the lower limit of experimentally determined values. We now widened up the discussion on page 18 lines 20-29 to include all potential reasons.

**• (Fig. 7 caption) As in Fig. 6 caption it should be stated that points correspond to AF 0.05 condition.**

Thank you for pointing this out, we have now updated the caption accordingly.

**• (Fig. 8) This figure seems a bit out of place and is of limited use in the explanation here. Perhaps it could be introduced earlier in the particle characterization section and returned to here?**

We have now moved to the beginning of the manuscript (new Fig.1) and introduce it on page 4 lines 6 and 20-21 by stating: "*The MCM-41 (see Fig. 1a and b) particles were synthesized following Beck et al., (1992),…*" and "*To obtain larger pore diameters (~9 nm), SBA-15 particles (see Fig. 1c and d) were synthesized similarly to Linton et al., (2009b) where Pluronic® P104 (1.25 g, BASF) was…*".

**• (25, 26) It might be a bit strong to say that parameterizations should be based on PCF. Perhaps, should include?**

We have reformulated the sentence to state: "… *parametrizations should include the PCF mechanism below the HFT*…" on page 27 lines 5-6.

**• (25, 26) The final sentence links this work to understanding anthropogenic emissions, but this is really the first mention of such emissions up to this point. Are these particles particularly analogous to any known anthropogenic emission? If the link is not strong I think this point can be left out, there are of course many reasons to better understand the effects of porosity and geometry on freezing.**

We have now added reference to soot particles as an example of anthropogenic emissions, which have recently been shown to nucleate ice following PCF to text as follows: "*…, such as soot, which has been shown to nucleate ice in accordance with PCF (Mahrt et al., 2018, 2020b, 2020a; Nichman et al., 2019),…*" on page 25 lines 13-15.

**Summary: I have enjoyed reading the submitted manuscript. I reiterate that I think it could benefit from an improved clarity with regard to the concrete conclusions the authors would like to posit. My understanding is that basic edifice of PCF which rests on the inverse Kelvin equation does a good job of predicting the experimental observations if some of the asserted assumptions are valid. It appears that as with many systems a complete understanding of the data would involve a much more comprehensive picture of the intermolecular interactions specific to each system. The measurements rely both on ice nucleation and crystal growth to a detectible size, thus many details related to both the ice initiation and macroscopic state are convoluted. Such complexities are intrinsic in many experiments, yet I still feel this study brings the community a step forward. However, it appears the open question remains as to how this might be utilized and tested in a messier real atmospheric aerosol scenario.**

We would like to thank the reviewer for commending the efforts made in this study to disentangle the complex relationship between pore size and ability of pores to uptake water within the PCF framework. Indeed, it is our intention that this study lays the foundation for future studies to further quantify the PCF mechanism for its application in real atmosphere particles thus improving the representation in cloud models via ice nucleation parametrizations.

**Reviewer References:**

**[1] Vali, G., DeMott, P. J., Mohler, O., and Whale, T. F. (2015). Technical note: A proposal for ice ¨ nucleation terminology. Atmospheric Chemistry and Physics, 15(18):10263–10270.**

**[2] Marcolli, C. (2014). Deposition nucleation viewed as homogeneous or immersion freezing in pores and cavities. Atmospheric Chemistry and Physics, 14(4):2071–2104.**

**[3] Bartels-Rausch, T., Jacobi, H.-W., Kahan, T. F., Thomas, J. L., Thomson, E. S., Abbatt, J. P. D., Ammann, M., Blackford, J. R., Bluhm, H., Boxe, C., Domine, F., Frey, M. M., Gladich, I., Guzman, ´ M. I., Heger, D., Huthwelker, T., Klan, P., Kuhs, W. F., Kuo, M. H., Maus, S., Moussa, S. G., McNeill, ´ V. F., Newberg, J. T., Pettersson, J. B. C., Roeselova, M., and Sodeau, J. R. (2014). A review of air– ´ ice chemical and physical interactions (AICI): liquids, quasi-liquids, and solids in snow. Atmospheric Chemistry and Physics, 14(3):1587–1633.**

**[4] Kong, X., Papagiannakopoulos, P., Thomson, E. S., Markovic, N., and Pettersson, J. B. C. (2014a). ´ Water accommodation and desorption kinetics on ice. The Journal of Physical Chemistry A, 118(22):3973–3979.**

**[5] Kong, X., Thomson, E. S., Papagiannakopoulos, P., Johansson, S. M., and Pettersson, J. B. C. (2014b). Water accommodation on ice and organic surfaces: Insights from environmental molecular beam experiments. The Journal of Physical Chemistry B, 118(47):13378–13386.**

**[6] Thomson, E. S., Kong, X., Markovic, N., Papagiannakopoulos, P., and Pettersson, J. B. C. (2013). ´ Collision dynamics and uptake of water on alcohol-covered ice. Atmospheric Chemistry and Physics, 13(4):2223–2233.**

**Author References:**

[revised manuscript text omitted]